# Revealing trends and persistent cycles of non-autonomous systems with autonomous operator-theoretic techniques

Gary Froyland [1] ✉, Dimitrios Giannakis [2,3], Edoardo Luna [4] & Joanna Slawinska [2]

An important problem in modern applied science is to characterize the behavior of systems with complex internal dynamics subjected to external forcings. Many existing approaches rely on ensembles to generate information from the external forcings, making them unsuitable to study natural systems where only a single realization is observed. A prominent example is climate dynamics, where an objective identification of signals in the observational record attributable to natural variability and climate change is crucial for making climate projections for the coming decades. Here, we show that operator-theoretic techniques previously developed to identify slowly decorrelating observables of autonomous dynamical systems provide a powerful means for identifying nonlinear trends and persistent cycles of non-autonomous systems using data from a *single* trajectory of the system. We apply our framework to real-world examples from climate dynamics: Variability of sea surface temperature over the industrial era and the mid-Pleistocene transition of Quaternary glaciation cycles.

Operator-theoretic techniques have proven to be highly successful at analyzing dynamical systems[1–3]. These techniques were primarily developed for autonomous dynamics, where the governing rules do not change over time. However, many important phenomena are influenced by changing external factors, resulting in time-dependent governing rules. Examples include collective motion of particles and organisms in response to changes in their environment[4,5], neuronal dynamics under stimuli[6,7], mixing and coherent structure formation under a time-dependent fluid flow[8,9], and the variability of the Earth's climate under natural and anthropogenic forcings[10,11]. In response, at the beginning of the previous decade, extensions of operator-theoretic techniques for non-autonomous time-asymptotic dynamics[12,13] were developed through transfer operator cocycles. Shortly after, techniques for handling non-autonomous finite-time dynamics were developed through singular vectors of transfer operators[14,15] and eigenvectors of the dynamic Laplacian[16,17]. These

mathematically rigorous methods enable the analysis of a much wider class of systems and only need a single forcing history, but like all truly non-autonomous methods of analysis, multiple trajectories or multiple observations along the single forcing history are used. On the Koopman operator side, two-parameter families of Koopman operators (estimated from multiple observations) have been studied for systems with periodic or quasiperiodic time-dependence[18]. Other approaches have developed extensions of the dynamic mode decomposition (DMD) technique[19,20] for autonomous systems that utilize time-dependent spectral computations on moving stencils[21], or external control inputs in the DMD operator estimation problem[22–24]. In control applications, one invariably has access to multiple *known* forcing histories for training.

In recent work[25], the authors developed a framework based on *autonomous* techniques for complex eigenvalues of transfer[26–28] and Koopman[2] operators and their corresponding complex eigenvectors

[1]School of Mathematics and Statistics, University of New South Wales, Sydney, NSW 2052, Australia. [2]Department of Mathematics, Dartmouth College, Hanover, NH 03755, USA. [3]Department of Physics and Astronomy, Dartmouth College, Hanover, NH 03755, USA. [4]Department of Physics, University of Texas at Austin, Austin, TX 78712, USA. ✉e-mail: g.froyland@unsw.edu.au

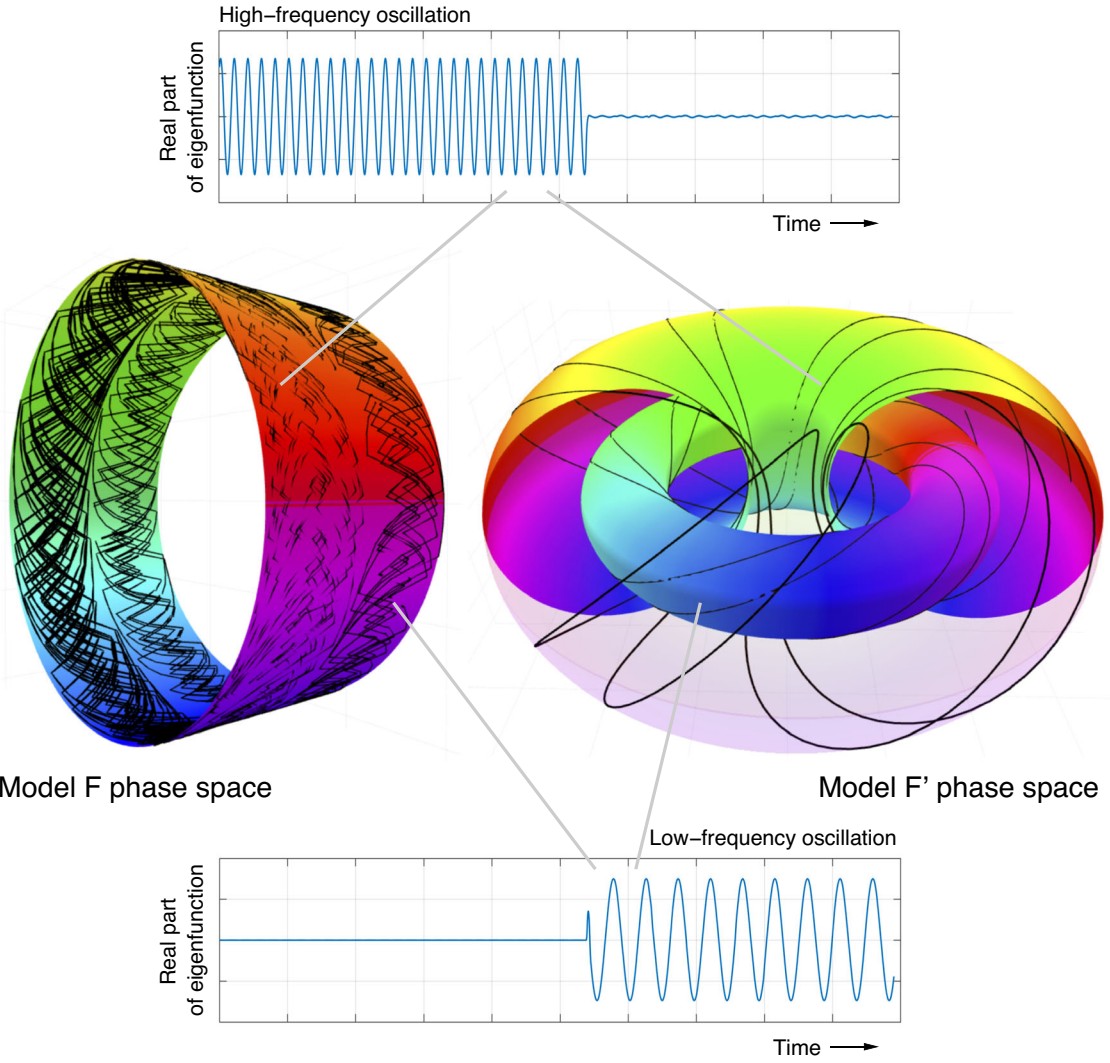

**Fig. 1 | Models of frequency switching (Model F) and co-existing frequencies (Model F′).** Center Left: The phase space $[0,1] \times S^1$ of Model F (see (2)) is identified with the surface of a cylinder. The (cyclic rainbow) colors represent the scalar output of the observation function $h(\theta) = \cos(\theta)$. The black line represents a trajectory that begins on the left half of the cylinder: the underlying frequency regime parameter $x \in [0,1]$ chaotically evolves on the left of the cylinder (corresponding to $x \in [0, 1/2]$), while the oscillation phase $\theta$ rotates around the cylinder with a fixed period of 40 time units. At some point, the frequency regime parameter $x$ switches to the right half of the cylinder (corresponding to $x \in [1/2, 1]$) and the trajectory thereafter proceeds with a constant rotation rate around the cylinder with a period of 97.35 time units. This switching models the change in glaciation cycle frequency from faster to slower, as observed after the MPT transition period (see section on Quaternary glaciation cycles). Upper: The real part of one of the complex eigenfunctions from the first complex-conjugate pair for Model F. The faster oscillation prior to the MPT is captured, while the amplitude of the slower post-MPT oscillation is suppressed. Lower: The real part of one of the complex eigenfunctions in the second complex-conjugate pair for Model F. The slower post-MPT oscillation is captured, while the amplitude of the faster pre-MPT oscillation is suppressed. Center Right: The domain $[0,1] \times S^1 \times S^1$ of model F′ (see (6)) is identified with a solid torus, where $x = 0$ corresponds to the outer toral shell and $x = 1$ corresponds to the inner toral shell. The (cyclic rainbow) colors represent the scalar output of the observation function $h$ in (7) on these shells. The black curve represents a trajectory of the system (6) with angular frequencies $\alpha_1 = 2\pi/40$ and $\alpha_2 = 2\pi/97.35$. The trajectory begins on the outer toral shell, where the recorded oscillation of color by the trajectory is relatively rapid. The trajectory then switches the the inner toral shell where the recorded oscillation of color is slower (by a factor $\alpha_1/\alpha_2$). This switching also models the change in glaciation cycle frequency from faster to slower, and additionally allows for superposition of two frequencies when the trajectory lies between the two toral shells (see equation (7)).

that successfully extracted slowly decaying (slowly decorrelating) *cycles* from a single time series. Specifically, the El-Niño Southern Oscillation (ENSO)[29] was extracted as a complex eigenvector of the transfer or Koopman operators built from monthly SST images over the past 50 years. Over this time, there is noticeable warming of the ocean, and one could argue that time-dependent techniques should be applied. Nevertheless, the extracted ENSO cycle was in excellent agreement with independent climate observations and displayed greater cyclicity (i.e., a well-defined characteristic frequency and slowly decaying correlation amplitude) than the corresponding ENSO cycle defined using a standard Niño 3.4 index.

We will show in this present work that an estimate of the (non-stationary) Indo-Pacific warming trend, as well as the modulation of the seasonal cycle by the trend, can be obtained through eigenvectors of transfer or Koopman operators when a small amount of diffusion is added. Here, by "trend" we mean a statistical trend arising from computations involving observations. A conceptual model of secular SST increase, combined with oscillatory behavior such as the seasonal cycle or ENSO, is simple harmonic motion with a drift in the mean. We also consider simple harmonic motion with a drift in the amplitude, and simple harmonic motion with a change in the frequency. We will use the latter as a toy model for the Mid-Pleistocene Transition (MPT)

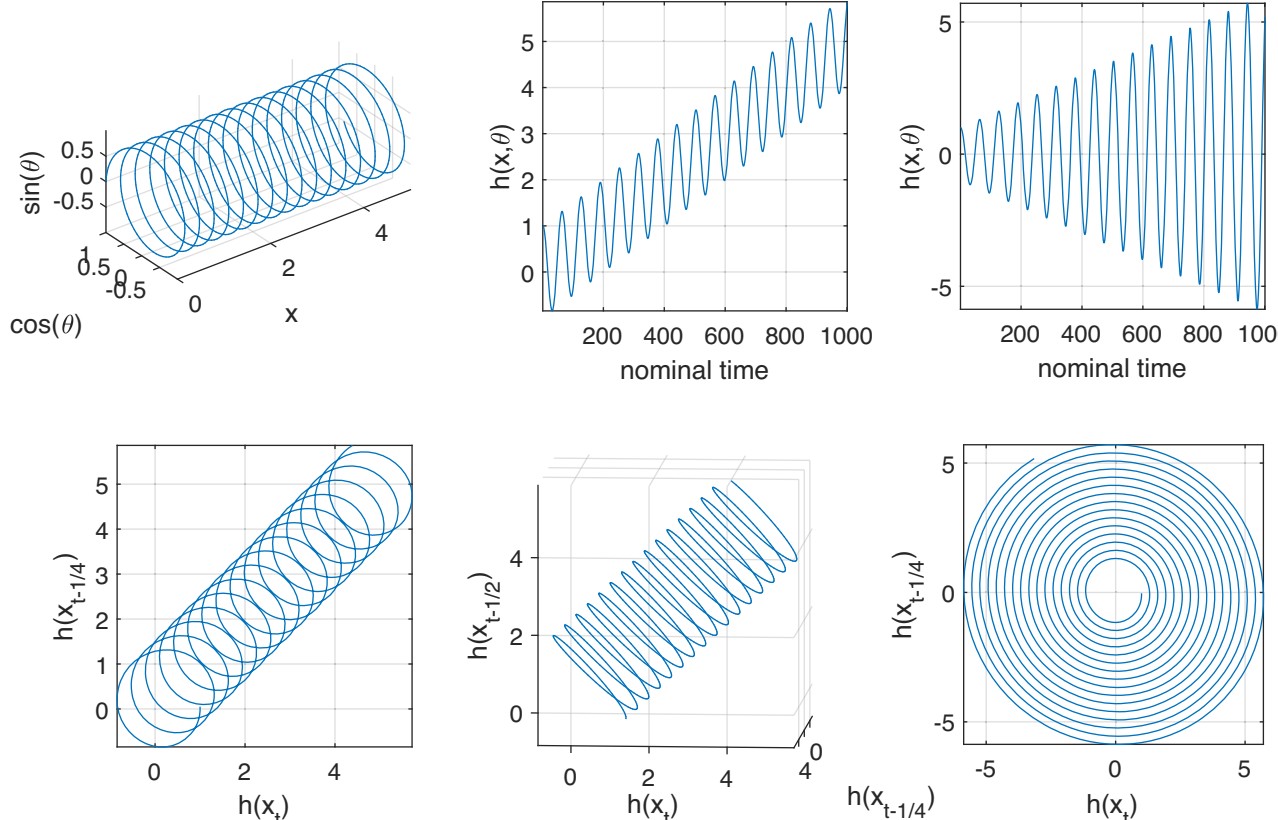

**Fig. 2 | Trajectories and embeddings for Models M and A.** Upper Left: A typical trajectory of the $(x, \theta)$ coordinates derived from iteration of $T$ in (1), with linear drift $d(t) = t/10$ and $\alpha = 1/10$, and $\tau = 100$. Upper Center: Graph of $h(x,\theta) = x + \cos(\theta)$ vs. time, illustrating the creation of an oscillatory signal with a linear drift in the mean of the signal. Upper Right: Graph of $h(x,\theta) = (1+x)\cos(\theta)$ vs. time, illustrating the creation of an oscillatory signal with a linear drift in the amplitude of the signal.

Lower Left: Two-dimensional time-delay embedding of the time series $h(x_t)$ shown in Fig. 2 (upper center), using a lag of $\ell = 1/4$ of the cycle period. Lower Center: Same as Left, but embedded in three dimensions. Lower Right: Two-dimensional time-delay embedding of the time series $h(x_t)$ shown in Fig. 2 (upper right), using a lag of $\ell = 1/4$ of the cycle period.

of Quaternary glaciation cycles[30], where we will show that transfer/Koopman operators successfully recover the pre- and post-transition cycles from field measurements of benthic $\delta^{18}O$ oxygen isotope ratio[31] (a paleo-proxy for ambient seawater temperature) as distinct eigenfunctions.

Each of the idealized models studied in this work is non-autonomous. Through these models, we provide theoretical explanations for why autonomous techniques can provide useful information from nonautonomous systems—see Fig. 1 for a schematic illustration of our approach applied to idealized frequency-switching models of the MPT. Our real-world results include reconstructions of surface air temperature (SAT) and precipitation fields using transfer/Koopman eigenfunctions computed from Indo-Pacific SST. These reconstructions reveal regions in South America that have undergone qualitative changes in the phasing and amplitude of the seasonal precipitation cycle over the industrial era. Furthermore, the eigenfunctions computed from $\delta^{18}O$ data identify the fundamental 40 kyr and 100 kyr Northern Hemisphere (NH) glaciation cycles over the past 3 Myr and the associated MPT. We find that the 40 kyr cycle and its harmonics persist after the MPT in a low-amplitude state, and the resulting interference with the 100 kyr mode helps explain variations in the amplitude and duration of post-MPT glaciation cycles.

## Results
### Model classes
Our two main real-world examples are drawn from two single time series: Indo-Pacific SST fields over the industrial era and benthic $\delta^{18}O$

records over the past 3 My. The SST time series incorporates many persistent co-existing cycles of differing frequencies in the sub-seasonal to interannual band, studied in[25], and a climate change trend, which we address in the present work. The $\delta^{18}O$ data has two main frequencies associated with Quaternary glaciation cycles that co-exist for some, but not all, of the time, as well as a nonstationary trend.

We develop idealized models that have these basic types of behavior, namely (i) oscillation with a drift in the mean of the oscillation; (ii) oscillation with a drift in the amplitude of the oscillation; and (iii) a switching between two different oscillations, possibly with co-existence. In each of our real-world examples, it is the underlying dynamics that is undergoing change (e.g., in response to changes in greenhouse gas forcings or orbital forcings of the climate), rather than the measuring device that records the time series. Therefore, we push the time-dependence of the time series into the dynamical system and keep the observation function time-independent.

### Model M: oscillation with a drift in the mean
A simple dynamical system that incorporates a cycle with a drift transverse to the cycle is helical dynamics around the curved surface of a cylinder. More precisely, we consider the (infinite) cylinder $\mathbb{R} \times S^1$, where $S^1$ is the circle with unit radius. We consider time-dependent dynamics represented by a discrete-time transformation $T : \mathbb{R}^+ \times \mathbb{R} \times S^1 \to \mathbb{R}^+ \times \mathbb{R} \times S^1$

$$T(t,x,\theta) = (t+1, x+d(t), \theta+\alpha), \tag{1}$$

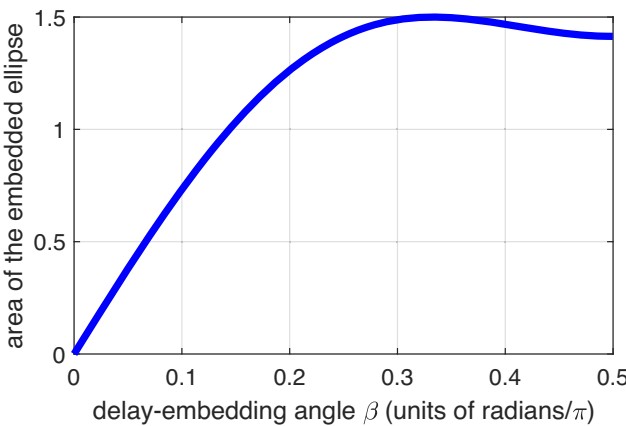

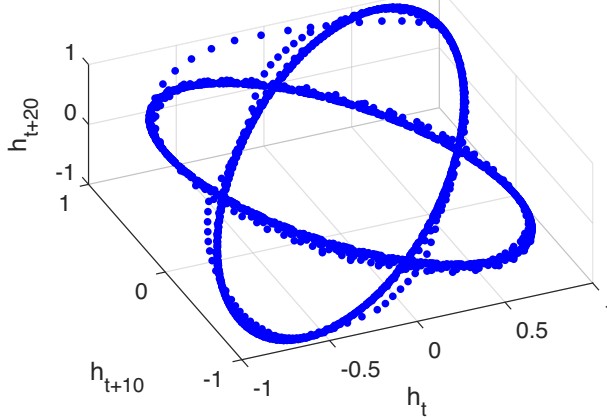

**Fig. 3 | Embedding multiple frequencies.** Left: Area of the embedded ellipse $(\cos(\theta), \cos(\theta+\beta), \cos(\theta+2\beta))$ as a function of $\beta$. *Right:* Embedding of the time series from Model F in Fig. 6 (right) with a lag $\ell$ of 10 time steps. One obtains two disjoint ellipses of large area that are well separated. The trace joining the ellipses occurs in the short regime where the time series switches between frequencies.

where $d(t)$ is a time-dependent drift along the central axis of the cylinder and $\alpha$ is the rate of rotation around the circular base of the cylinder. In this model the coordinate $t$, representing time, is the driving coordinate. Iterating $T$ for $\tau$ iterations, and plotting trajectories of the $(x, \theta)$ coordinates on the cylinder embedded in $\mathbb{R}^3$, we obtain a picture such as in Fig. 2 (upper left).

A continuous-time representation of this dynamics is given by the vector field $F : \mathbb{R}^+ \times \mathbb{R} \times S^1 \to \mathbb{R}^3$ defined by $F(t, x, \theta) = (1, d(t), \alpha)$. To create a scalar time series of oscillation with drift, we observe with the time-independent function $h(x, \theta) = x + \cos\theta$, which additively combines the phase of the oscillation $\theta$ with the current drift state $x$ to create an oscillation with a drift; see Fig. 2 (upper center).

### Model A: oscillation with a drift in the amplitude
The same dynamical model $T$ as in (1) can be used with the altered observation function $h(x, \theta) = (a + x)\cos\theta$, to create an oscillatory time series with an amplitude that varies from $a$ according to the current drift state $x$. See Fig. 2 (upper right).

### Model F: switching between two different oscillation frequencies
We create a simple class of time-dependent dynamics that can model switching between two distinct oscillations. The state variable $x$ that previously represented the state of the drift will now represent a frequency "regime". This frequency-regime coordinate $x$ is the driving coordinate and evolves according to metastable dynamics, for example according to $f_\delta : [0, 1] \to [0, 1]$, for small $\delta > 0$, where

$$f_\delta(x) = \begin{cases} 2x, & 0 \leq x < 1/4, \\ \delta + 2(x - 1/4) \pmod 1, & 1/4 \leq x < 3/4, \\ 1/2 + 2(x - 3/4), & 3/4 \leq x \leq 1. \end{cases}$$

The map $f_\delta$ is a chaotic map that preserves Lebesgue measure on $[0, 1]$. Each of the intervals $[0, 1/2]$ and $[1/2, 1]$ is almost-invariant, and the average probability to switch between these intervals at each iteration is $\delta$. The two intervals will represent two distinct frequency regimes in the angular coordinate introduced below. We consider discrete-time dynamics $T : [0, 1] \times S^1 \to [0, 1] \times S^1$ defined by

$$T(x, \theta) = (f(x), \theta + w(x)\alpha_1 + (1 - w(x))\alpha_2), \quad (2)$$

where $w : [0, 1] \to [0, 1]$ is a switching function defined by $w(x) = (1 + \tanh(c \cdot (x - 1/2)))/2$, and $c > 0$ is a parameter. For $c$ large (we set $c = 40$ in all of our experiments) $w(x)$ takes values approximately equal to zero on the interval $([0, 1/2)$, and $w(x)$ takes values approximately to 1 on the interval $(1/2, 1])$. Thus, the angular coordinate $\theta$ in (2) advances by either $\alpha_1$ or $\alpha_2$, depending on the underlying frequency regime controlled by $x$. Figure 1 (left) illustrates an orbit on the cylinder for the map $T$; one sees faster rotation around one half of the cylinder in comparison to the other half. To create a time series that switches between the two frequencies, we simply use the (driving-independent) observation function $h(\theta) = \cos\theta$, which records the current phase of the oscillation; see Fig. 6 (upper left).

### Why are the outputs of these models idealizations of our data?
**Indo-Pacific SST over the industrial era.** For Models A and M we imagine that $t$ is time, that $x$ is a proxy variable for the degrees of freedom of the climate system affected by greenhouse gas forcings, and that $\theta$ is the phase of an oscillatory process such as the seasonal cycle or ENSO. The observation function $h$ is an aggregate quantity (e.g., globally averaged SAT or SST).

**Benthic $\delta^{18}$O records over the Pleistocene.** For Model F we imagine that $x$ is a proxy variable for drivers of global mean temperature, and that $\theta$ is the phase of glaciation cycles. Later on we will develop Model F' (see Methods) as an alternative to Model F that is based on two phase angles: $\theta_1$ is the phase of the axial tilt cycle and $\theta_2$ is the phase of the orbital procession/inclination. For both Models F and F' the observation function $h$ outputs the $\delta^{18}$O isotope ratio (which correlates with global mean temperature).

### Time-delay embedding
In practice we imagine that we only have access to a single time series, and that this time series is generated by nonautonomous dynamics with a fixed observation function. A classical approach to such a time series might be to embed it in a higher-dimensional space using Takens' method of delays[32–34]. In this approach, observations $h_t$ with values in $\mathbb{R}^d$ are boosted in dimension through temporal concatenation to produce a new time series $\mathbf{h}_t := (h_t, h_{t-\ell}, h_{t-2\ell}, \ldots, h_{t-(Q-1)\cdot\ell}) \in \mathbb{R}^{dQ}$, where $\ell$ is a *lag* and $Q$ is the number of lags. By various types of delay-embedding theorems valid for autonomous[35–37] or nonautonomous[38,39] dynamics, the delay-embedded data can, under appropriate assumptions, faithfully represent the underlying system state even if the observation map $h$ giving the time series data is non-injective.

Yet, even though the original dynamics can be theoretically recovered from observational data, identification of drifts remains notoriously difficult in practice. In this work, we demonstrate

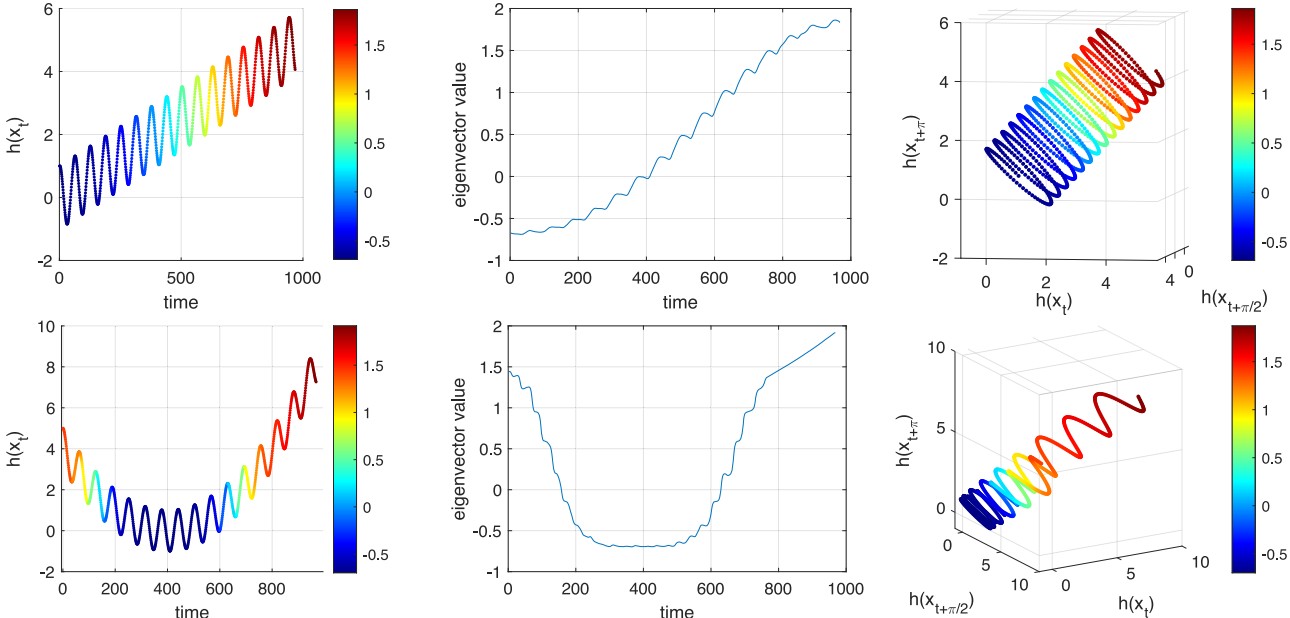

**Fig. 4 | Trend in the mean the signal from Model M is identified by the first nontrivial real eigenvector of the approximate transfer operator *P*.** Upper: Linear drift. The left panel shows the raw time series vs. time, colored according to the first nontrivial real eigenvector (second in the global ordering $v_2$); center panel shows the second eigenvector value vs. time; right panel shows the embedded data colored according to $v_2$. Lower: Non-monotonic quadratic drift. The left, center, and right panels show analogous objects as for linear drift.

that autonomous operator-theoretic methods coupled with delay-embedding can yield useful analyses of nonstationary dynamics. We proceed through the three model classes we have set up in the previous section.

**Delay embedding with a drift in the mean and amplitude – Models M & A.** Our underlying dynamics (1) is occurring on a helix embedded in $\mathbb{R}^3$ (Fig. 2 (upper left)). Thus, using an embedding lag $\ell$ of 1/4 of the oscillation period, we expect to require at least two embedding dimensions for the scalar signal of $h$ to accurately reconstruct the two-dimensional phase space. Figure 2 (lower left) shows that when the observation $h$ incorporates a drift in the mean, two dimensions are insufficient to uniquely recover the system state; three dimensions (lower center) are sufficient. When the observation function only has drift in the amplitude, two embedding dimensions are sufficient to recover a two-dimensional surface; see Fig. 2 (lower right). Note that the annulus shown in Fig. 2 (lower right) is diffeomorphic to a cylinder.

**Delay embedding with oscillation frequency switch – Model F.** The switching dynamics given by (2) take place on a cylinder; see Fig. 1 (left). To correctly recover the dynamics we require three embedding dimensions. In the previous subsection, we chose an embedding lag $\ell$ of 1/4 of the oscillation period. In Model F we have two distinct frequencies, which appear in bursts. Below we will use an abstract result (Lemma 1) to show that with a fixed lag, bursts of *any number of frequencies* may be successfully embedded in three dimensions. This result will also indicate how to best choose a single lag.

Consider a unit circle in the plane, centered at the origin; it has a parametric representation $(x,y) = (\cos(\theta), \sin(\theta)) \in \mathbb{R}^2$. We embed this circle with lagged observation $\cos(\theta)$; i.e., we consider the parametric curve $(\cos(\theta), \cos(\theta + \beta), \cos(\theta + 2\beta))$ in $\mathbb{R}^3$ for some lag $\beta$ and with $\theta \in (0, 2\pi)$. We then have:

**Lemma 1.** For each $\beta \in \mathbb{R}$, the parametric curve

$$\gamma_\beta(\theta) := (\cos(\theta), \cos(\theta + \beta), \cos(\theta + 2\beta)), \quad \theta \in [0, 2\pi), \tag{3}$$

is an ellipse embedded in $\mathbb{R}^3$. These ellipses are disjoint for distinct $\beta \in (0, \pi/2)$. When $\beta = \pi/3$ the curves are embedded circles, and the area inside the curves is maximized.

The proof of Lemma 1 is in Methods. Here, we show that the set $\{\gamma_\beta(\theta): 0 \le \theta < 2\pi\}$ is an ellipse for each $\beta$. Note that the coordinates of $\gamma_\beta(\theta)$ can be written as

$$
\begin{aligned}
&(\cos(\theta), \cos(\theta + \beta), \cos(\theta + 2\beta)) \\
={}&(\cos(\theta), \cos(\beta)\cos(\theta) - \sin(\beta)\sin(\theta), \\
&\quad \cos(2\beta)\cos(\theta) - \sin(2\beta)\sin(\theta)) \\
={}&(x, \cos(\beta)x - \sin(\beta)y, \cos(2\beta)x - \sin(2\beta)y),
\end{aligned}
$$

where $x = \cos\theta$ and $y = \sin\theta$. Thus, our delay embedding can be considered as mapping the unit circle linearly via $\Phi_\beta : \mathbb{R}^2 \to \mathbb{R}^3$, where

$$\Phi_\beta(x,y) = (x, \cos(\beta)x - \sin(\beta)y, \cos(2\beta)x - \sin(2\beta)y).$$

By linearity of $\Phi_\beta$, the unit circle in $\mathbb{R}^2$ will be transformed to an ellipse embedded in $\mathbb{R}^3$. Figure 3 (left) displays the area of the embedded ellipse as a function of $\beta$. The region between $\beta = \pi/4$ and $\beta = \pi/2$ is a relative "sweet spot".

We now apply Lemma 1 to our time series with frequency switching generated by Model F. We have two rates of rotation $\alpha_1$ and $\alpha_2$, and we therefore generate bursts of samples of single frequencies that look like $\theta = j\alpha_1$ and $\theta = j\alpha_2$, where $j = 0, 1, 2, \ldots$. Suppose that we embedded these bursts with the same lag $\ell$ in three dimensions; then we obtain chunks of embedding coordinates of the form: $(\cos(j\alpha_1), \cos((j+\ell)\alpha_1), \cos((j+2\ell)\alpha_1))$ and $(\cos(j\alpha_2), \cos((j+\ell)\alpha_2), \cos((j+2\ell)\alpha_2))$. We may apply Lemma 1 with $\beta_1 = \ell\alpha_1$ and $\beta_2 = \ell\alpha_2$, to immediately see that these discretely sampled simple harmonic motions with differing frequencies embed in $\mathbb{R}^3$ as *disjoint ellipses*, provided that $0 < \beta_1 \ne \beta_2 \le \pi/2$. In our example we have $\alpha_1 = 2\pi/40 > \alpha_2 = 2\pi/97.35$, so we set $\ell = 10 \approx (\pi/2)/(2\pi/40)$ in order that the larger angle $\beta_1$ hits the outer range of the interval $(0, \pi/2)$. With $\ell = 10$, we then have $\beta_2 = \ell\alpha_2 \approx 0.645$, which yields a large area for the

embedded ellipse, as seen in Fig. 3 (left). Figure 3 (right) shows this embedding.

## Transfer and Koopman operator computation

Given a dynamical system $T:\Omega\to\Omega$ on a domain $\Omega$ and a complex-valued function $f:\Omega\to\mathbb{C}$, the transfer operator $\mathscr{L}$ is defined by $\mathscr{L}f=f\circ T^{-1}$, where $T^{-1}$ may be multivalued if $T$ is non-injective. The transfer operator is the natural pushforward action on functions. The Koopman operator is defined as $\mathscr{K}f=f\circ T$, and is the natural pullback on functions. We now describe the construction of transfer operators directly from the embedded time series data. While the approach to numerically constructing the transfer operator is a slight modification of the methodology in[25], we provide a new interpretation of this numerical approximation to explain the form of the eigenfunctions in Models M and A. The time-delay embedding of Models M, A, and F uses scalar outputs $h(x,\theta)\in\mathbb{R}$. Later we will discuss time series observations of SST and $\delta^{18}$O. In the former situation our observations $h$ are vector valued and lie in $\mathbb{R}^d$. As in the case of scalar-valued observations, we time-delay embed these vector-valued observations by concatenation. We obtain a time series of embedded data: $\mathbf{h}_t:=(h_t,h_{t-\ell},h_{t-2\ell},\ldots,h_{t-(Q-1)\cdot\ell})\in\mathbb{R}^{dQ}$ for $t=1,\ldots,N$. For models M, A, and F, we have $d=1$ and $Q=3$ and so $\mathbf{h}_t:=(h_t,h_{t-\ell},h_{t-2\ell})\in\mathbb{R}^3$. We select a forward-step time $s>0$ (typically corresponding to a single sampling index) and construct a square array

$$S_{ij}=\exp\left(-\frac{\|\mathbf{h}_i-\mathbf{h}_{j+s}\|^2}{d_id_{j+s}}\right),\quad i,j=1,\ldots,N-s, \tag{4}$$

where $d_i$ is the distance from $\mathbf{h}_i$ to its $K^{\text{th}}$ nearest neighbor for modest $K$. Selecting $K\approx 7$ usually suffices unless it is numerically challenging to compute eigenvectors, in which case $K$ could be increased. We then row-normalize the matrix $S$ to obtain a row-stochastic Markov matrix

$$P_{ij}=S_{ij}/\sum_j S_{ij},\quad i=1,\ldots,N-s. \tag{5}$$

Note that the diffusion mentioned in the Introduction has been added through the use of the kernel $S$ on finite data. If we think of a vector $\mathbf{f}\in\mathbb{R}^{N-s}$ as values of a function $f:\Omega\to\mathbb{C}$ along our trajectory then the vector $P\mathbf{f}$ corresponds to pushing $f$ forward $s$ steps by $\mathscr{L}^s f$. We will compute eigenvectors $v$ such that $Pv=\lambda v$; in particular, multiplication on the right corresponds to forward evolution in time. We note that by construction $P\mathbf{1}=\mathbf{1}$ (where $\mathbf{1}$ is the $(N-s)$-vector with all entries equal to 1) and thus the leading eigenvalue is 1 and the corresponding eigenvector is constant. We will typically be concerned with the second and lower eigenvalues and eigenvectors. If we wish to instead approximate the Koopman operator, we can choose a backward-step time $s<0$.

Note that the matrix $S$ in (4) can be viewed as a non-symmetric discrete heat kernel or non-symmetric diffusion matrix, and the matrix $P$ as the corresponding Markov process. This non-symmetry will be small provided that the step length $s$ is small relative to the length $N$ of the time series. If the bandwidths $d_i$ in (4) are chosen so that the resulting diffusion is commensurate with the advection in embedding space due to the time step $s$ we can expect leading real eigenvalues of $P$ (due primarily to diffusion) to be mixed with leading complex eigenvalues of $P$ (due to periodicities arising from the advective components). The diffusive aspects of the matrix $P$ will be important for the following theoretical explanations of the forms of the operator eigenfunctions for our idealized models.

## Eigenfunctions for Model M

Figure 2 (upper center) showed an oscillatory time series with a linear drift in the mean of the oscillation. Figure 2 (lower center) showed a three-dimensional embedding of these observations, reproducing a diffeomorphic copy of the original dynamical system shown in Fig. 2

(upper left). We now build a transfer operator approximation on the embedded time series data in Fig. 2 (lower center), according to equations (4) and (5) with $s=1$ and $K=7$. Figure 4 (upper row) displays different views of the second eigenvector $v_2$ with eigenvalue $\lambda_2=0.9943$. On the left is the original time series, colored by the value of $v_2$. In the center is the value of $v_{2,t}$ plotted against time index $t$. The key point is that we see qualitative agreement in the behavior of the eigenvector value versus time (center) and the mean value of the drift (left). At the right is the embedded time series, colored by the value of $v_2$.

We remarked above that the matrix $P$ can be seen as an approximation of a slightly biased Markov diffusion process on the spiral structure. The slight bias is due to the the forward-step time $s=1$ rather than $s=0$, which would be unbiased. The second eigenvector of such a diffusion process is consistent with the upper-right panel of Fig. 4. Thus the underlying reason why Fig. 4 is colored from one end of the helix to the other is that $P$ represents a slightly biased random walk along the one-dimensional spiral.

Focusing now on the lower row of Fig. 4, in the left panel we construct an asymmetric quadratic drift in the mean of the oscillation. We compute $v_2$ (with eigenvalue $\lambda_2=0.9984$) using the corresponding three-dimensional embedding shown in the lower-right panel as the basis for constructing the matrix $P$. In the center panel of the lower row of Fig. 4 we see that the eigenvector value versus time qualitatively reflects the drift in the mean of the oscillatory signal in the left panel. The reason for this correspondence is the same as for the linear drift, but slightly more complicated because of the non-monotonicity of the drift. In the embedded space in the lower-right panel of Fig. 4 we see that the embedded signal doubles back over itself. We may again consider $P$ as approximating a slightly biased Markov diffusion process on the structure in the lower-right panel of Fig. 4. The eigenvector values appear as they do because of the geometric proximity of different parts of the embedded time series.

We recall that the choice of (forward) time step $s=1$ corresponds to $P$ approximating a transfer operator, but we could also have chosen $s=-1$ so that $P$ approximates a Koopman operator and the above results would be similar. This is also the case for the following experiments with Models A and F.

## Eigenfunctions for Model A

In the upper row of Fig. 5 we show (left to right): (i) a time series with linear drift in the amplitude; (ii) the values of the leading nontrivial real eigenvector ($v_4$) versus time; and (iii) the two-dimensional embedded time series colored according to the values of $v_4$. All computations were made with $s=1$ and $K=7$. The second and third eigenvalues are a complex-conjugate pair $0.9891\pm0.0991i$ corresponding to a rotation of 0.0998 radians per time step, which is almost exactly the rotation rate of $\alpha=1/10$ noted in Fig. 2. We are interested in extracting the drift in the amplitude and we move on to the first nontrivial real eigenvalue, which appears in position 4: $\lambda_4=0.9825$. This eigenvalue is real because there is no oscillation associated with the drift in the amplitude. The embedded points in the upper-right panel of Fig. 5 are colored from "inside to out" because we may again interpret $P$ as a slightly biased approximate Markov diffusion process. In particular, the upper-center panel qualitatively extracts the increasing amplitude.

The lower-left panel of Fig. 5 shows a more complicated non-monotonic quadratic amplitude variation. This variation is captured in the lower-center panel from the leading nontrivial real eigenvector $v_6$, for the same reasons as those given above, namely the geometry of the embedding drives the form of the eigenvector.

## Eigenfunctions for Model F

We return to the time series in the upper-left panel of Fig. 6. This time series is embedded as in Fig. 3 (right), leading to two ellipses that are disconnected, apart from the short trajectory joining them. We

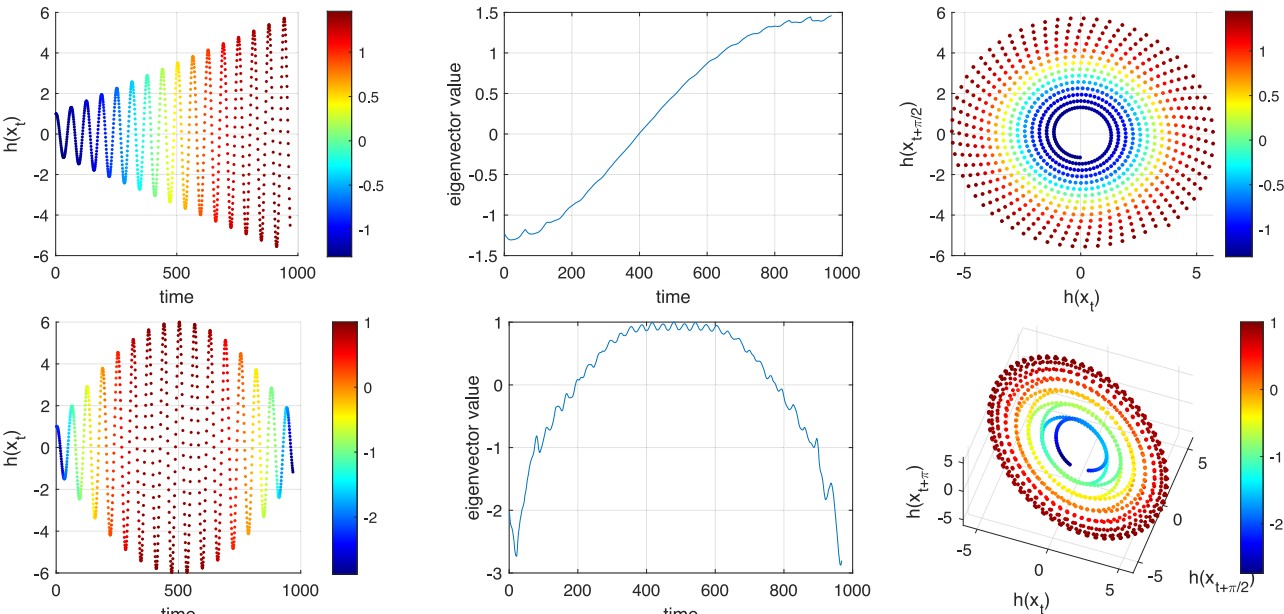

**Fig. 5 | Trend in the amplitude of the signal from Model A is identified by the first nontrivial real eigenvector of the approximate transfer operator *P*.** Upper: Linear drift. The left panel shows the raw time series vs. time, colored according to the first nontrivial real eigenvector (fourth in the global ordering $v_4$); center panel shows the second eigenvector value vs. time; right panel shows the embedded data colored according to $v_4$. Lower: Non-monotonic quadratic drift. The left, center, and right panels show analogous objects as for linear drift. The vector shown is again the first nontrivial real eigenvector (sixth in the global ordering $v_6$).

construct the matrix $P$ as above with $s = 1$, but we increase $K$ to 25 in order to achieve a robust numerical solution for the eigenvectors. We find that eigenvalues 2 and 3 are a complex-conjugate pair with $\lambda_2 = 0.9866 + 0.1547i$ and eigenvalues 4 and 5 are another complex-conjugate pair with $\lambda_4 = 0.9954 + 0.0660i$. The arguments of these complex eigenvalues correspond to rotation periods of 40.39 and 94.95 time units, which compare to exact values of 40 and 97.3537. The corresponding eigenvectors are shown in Fig. 6.

In the left panels of Fig. 6 we see the original time series (upper), the real parts of one of the eigenvectors from the leading complex pair versus time (center), and the real parts of one of the eigenvectors from the second complex pair versus time (lower). The embedded time series data colored according to the real parts of the two complex eigenvectors is shown in the right panels. In particular, we note that the oscillation period in the first half of the center left panel is identical to the oscillation period in the first half of the upper-left panel (the original time series), while the amplitude in the second half of the time series is suppressed in the eigenvector values (center left). In this sense, the leading complex eigenvector pair has *extracted the cycle that is supported on the first half of the time series.*

Similarly, in the lower-left panel of Fig. 6 we see that the large amplitude is concentrated in the second half of the time span and that the oscillation period is nearly identical to the oscillation period of the corresponding second half of the original time series (upper left). This second half of the eigenvector corresponds to one of the two colored ellipses in the right-most panel. Thus, the second complex eigenvector pair has extracted the second frequency in the time series. Overall, the two leading complex eigenvector pairs have *identified and filtered the distinct frequencies of the original signal.*

The reason why the eigenvectors of $P$ behave in this way is analogous to our discussion with the drift in the mean and amplitude. The geometry of the embedded time series again plays a key role. In this case, we have two ellipses that – apart from some small regions – are disconnected from one another. If we had two completely disconnected ellipses we should obtain two complex-conjugate eigenvalue pairs describing the rotation frequency for each ellipse; see[25] for

details. In the present setting, the embedded time series is slightly perturbed from this idealized situation, and we therefore obtain eigenvectors that are a slight perturbation of the eigenvectors in the idealized situation.

A frequency-switching model has been analysed using DMD with moving time windows, which are chosen so as to not overlap a switching time[21]. We note that our theory and numerical approach recovers the eigenvalues and eigenfunctions associated with the two frequency regimes of Model F by eigendecomposition of a single matrix $P$, without making use of moving windows or windowing to subdivide the time series into distinct frequency regimes.

## Climate variability and trends over the industrial era

One of the key challenges in advancing our scientific understanding of climate dynamics and improving the skill of climate forecasts and projections is to objectively identify the fundamental modes of climate variability, operating on timescales spanning months (seasonal cycle) to decades (low-frequency oceanic variability) under the influence of time-dependent natural and anthropogenic external forcings[40–42]. When analyzing observational data, we sample a single dynamical trajectory through observational networks with time-dependent biases, making the task of delineating natural variability from forced response particularly challenging[43,44].

As noted in the Introduction, previous work[25] has shown that operator-theoretic techniques can successfully extract fundamental cycles of climate dynamics such as ENSO, the seasonal cycle, and combination modes between ENSO and the seasonal cycle[45] from monthly-averaged SST snapshots, *despite the presence of a nonstationary climate-change trend in the data and the fact that the analysis techniques were originally designed for autonomous dynamics.* In addition, the transfer/Koopman operator spectral decompositions yield eigenfunctions with nonstationary associated time series that are broadly consistent with accepted climate change signals over the satellite era[46]. The spectra also contain "trend combination modes" corresponding to products between the trend and seasonal cycle eigenfunctions. Building upon the idealized models described in the

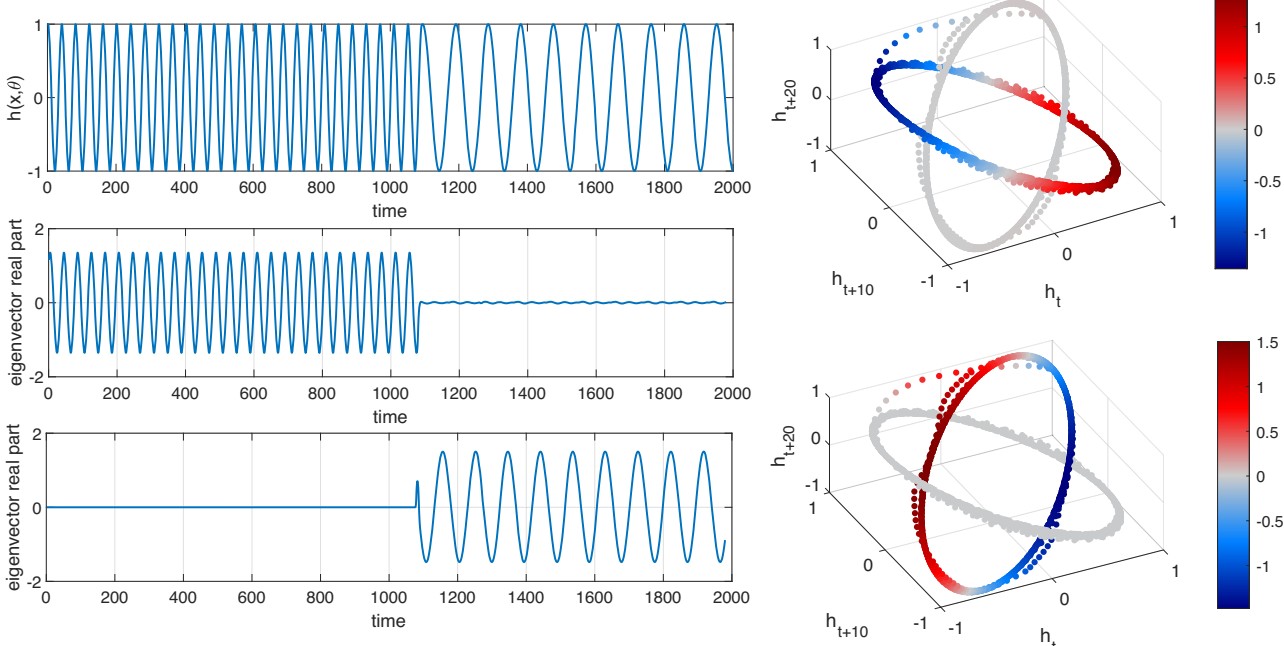

**Fig. 6 | Separation of the two distinct frequencies in Model F by two complex eigenvectors from distinct complex-conjugate pairs of the approximate transfer operator $P$.** Model F is defined in (2) with $\delta = 7.5 \times 10^{-4}$, $\alpha_1 = 1/40$, and $\alpha_2 = 1/97.3537 \approx 1/100$. Upper Left: Graph of observation function $h(x,\theta) = \cos(2\pi\theta)$ vs. time, illustrating the creation of an oscillatory signal switching between two frequencies. Center Left: The real part of an eigenvector from the first complex pair vs. time. Lower Left: The real part of an eigenvector from the second complex pair vs. time. Upper Right: The three-dimensional embedding of the time series, colored by the real part of an eigenvector from the first complex pair. Lower right: As for the upper right, but with the second complex eigenvector pair.

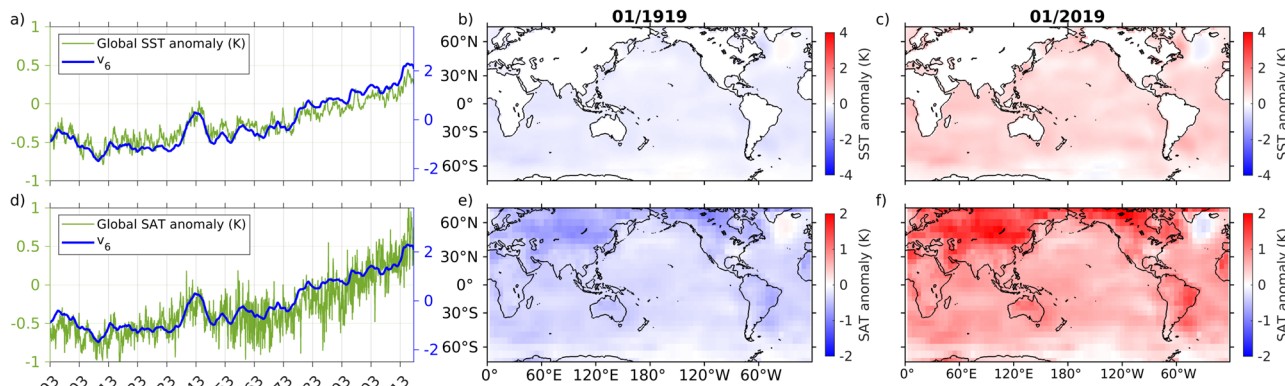

**Fig. 7 | Reconstruction of SST and SAT trends over the industrial era using eigenfunction $v_6$ of the generator.** Panels (**a**, **d**) show time series of globally averaged SST and SAT anomalies, respectively, along with time series of $v_6$. Panels (**b**, **c**) show global spatial maps of SST anomalies for January 1919 and 2019, respectively. Panels (**e**, **f**) show global SAT anomalies for the same dates.

preceding sections, in this section we extend the analysis of ref. 25 to an interval spanning the past ~130 years of the industrial era.

Regarding prior work, a great variety of mathematical and numerical techniques have been developed for analysis and modeling of climate dynamics[47,48]. Some approaches[49,50] employ state-space techniques for non-autonomous dynamical systems, such as the theory of pullback attractors, to characterize changes to natural variability under time-dependent climate forcings. On the data analysis side, Singular Spectrum Analysis[34,51] (SSA) and its equivalent Extended Empirical Orthogonal Function (EOF) analysis[52], combine aspects of state-space reconstruction using delay-embedding maps with eigendecomposition of covariance operators to extract oscillatory components and trends from climatic time series with higher fidelity than

standard EOF techniques. While SSA targets high-variance modes, the transfer/Koopman operator approach targets eigenfunctions with slow correlation decay. Another difference is that transfer/Koopman operator spectra provide direct estimates of characteristic oscillatory timescales, whereas in SSA the assignment of characteristic timescales requires post-processing of the eigenfunctions such as Fourier analysis. In the asymptotic limit of infinite delays, the SSA eigenspaces corresponding to nonzero eigenvalues (i.e., nonzero explained variance) are finite orthogonal direct sums of Koopman eigenspaces[53,54]. This provides an approximate interpretation of the cyclicity of the dominant modes extracted by SSA using many delays. Besides diagnostic applications such as the study presented here, operator-theoretic methods have been successfully employed in parametric

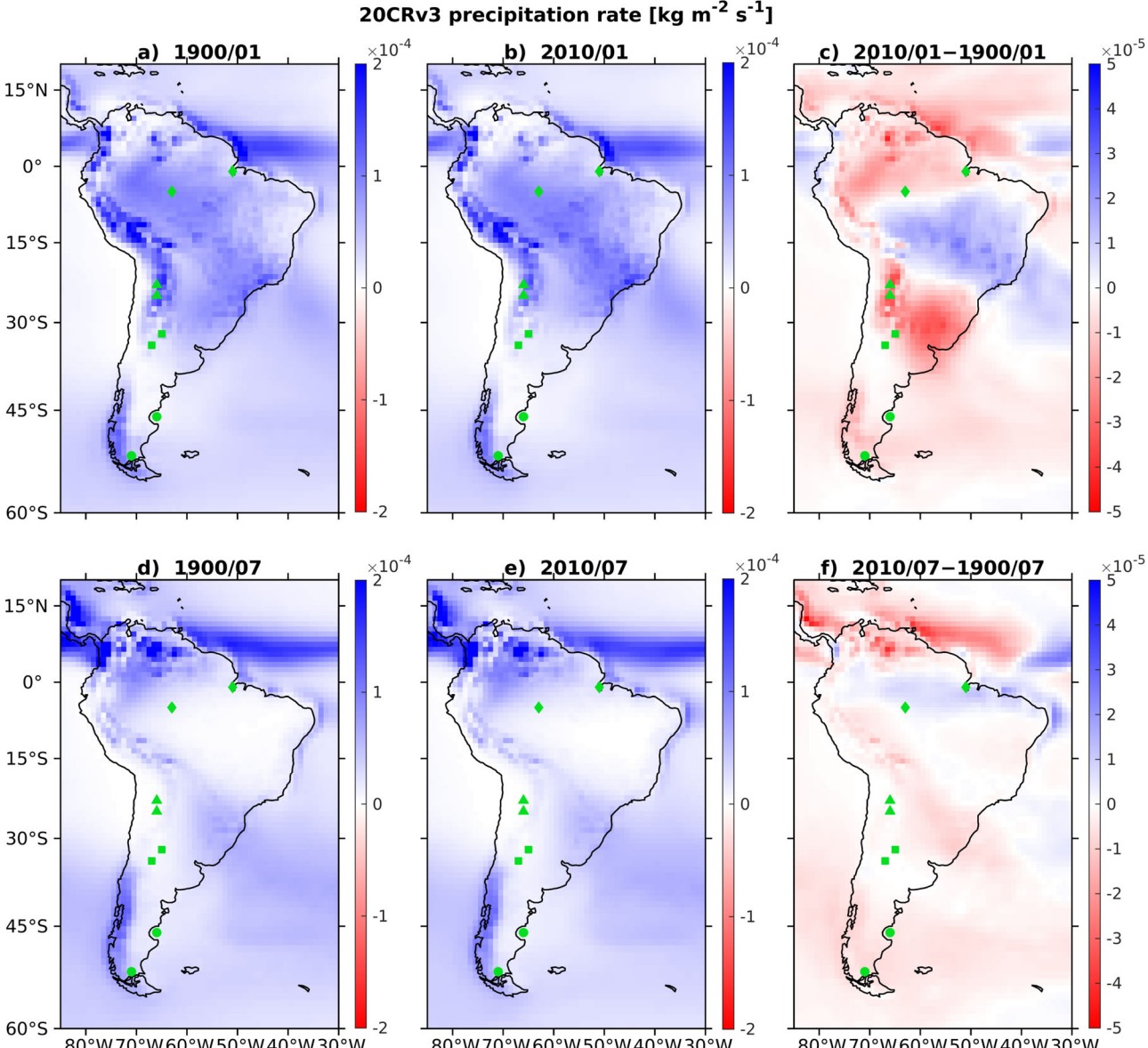

**Fig. 8 | Spatial maps of reconstructed 20CRv3 precipitation rate over South America using eigenfunctions representing the global climatology, seasonal cycle, trend, and products between seasonal cycle and trend.** Panels (**a**, **b**) show reconstructions of precipitation (in kg m⁻² s⁻¹) for January 1900 and 2010, respectively. Panels (**d**, **e**) show reconstructions for July of the same years, respectively. Panel (**c**) shows the difference between (**b**) and (**a**), and Panel (**f**) the difference between (**e**) and (**d**). The reconstructions in Panels (**c**, **f**) provide an estimate of the change in the seasonal cycle of precipitation over the industrial era. Markers in the shapes of triangles, diamonds, squares, and circles indicate the spatial locations sampled in Fig. 9a–d, respectively.

and non-parametric prediction models of climate phenomena, including ENSO[55,56], Pacific SST variability[57], tropical intraseasonal oscillations[58,59], and sea ice cover[60,61].

We analyze monthly-averaged SST fields with a 2° resolution from the ERSSTv4 reanalysis product[62] on the Indo-Pacific domain 28°E–70°W, 60°S–20°N. In addition, we analyze monthly-averaged, 5° global SAT anomaly data from the NOAA Global Surface Temperature Dataset (NOAAGlobalTemp), Version 5.0[63], and monthly-averaged, 1° global precipitation data from the NOAA/CIRES/DOE 20th Century Reanalysis version 3 (20CRv3)[64]. This analysis interval is longer than the 1970–2019 interval studied in ref. 25 as the focus of this paper is on long-term climate change trends which are more significant over our analysis period that covers a significant portion of the industrial era.

We extract eigenfunctions $v_j$ from the Indo-Pacific SST data using the kernel-based approach for approximating the generator of the Koopman/transfer operator groups described in ref. 25. Note that the SAT and precipitation fields, as well as SST fields outside our Indo-Pacific domain, are not employed in eigenfunction computations—we use these fields instead as target variables for reconstruction based on our eigenfunctions. Additional details on the operator and reconstruction calculations are provided in Methods. Supplementary Fig. 1 displays the spectrum of the generator, where the real and imaginary parts of the eigenvalues represent growth rate and oscillatory frequency, respectively. The eigenvalues can be grouped into families associated with the seasonal cycle, ENSO, trends, and decadal variability, similarly to ref. 25. In this paper, we will focus on the trend and seasonal-cycle modes. Note that each

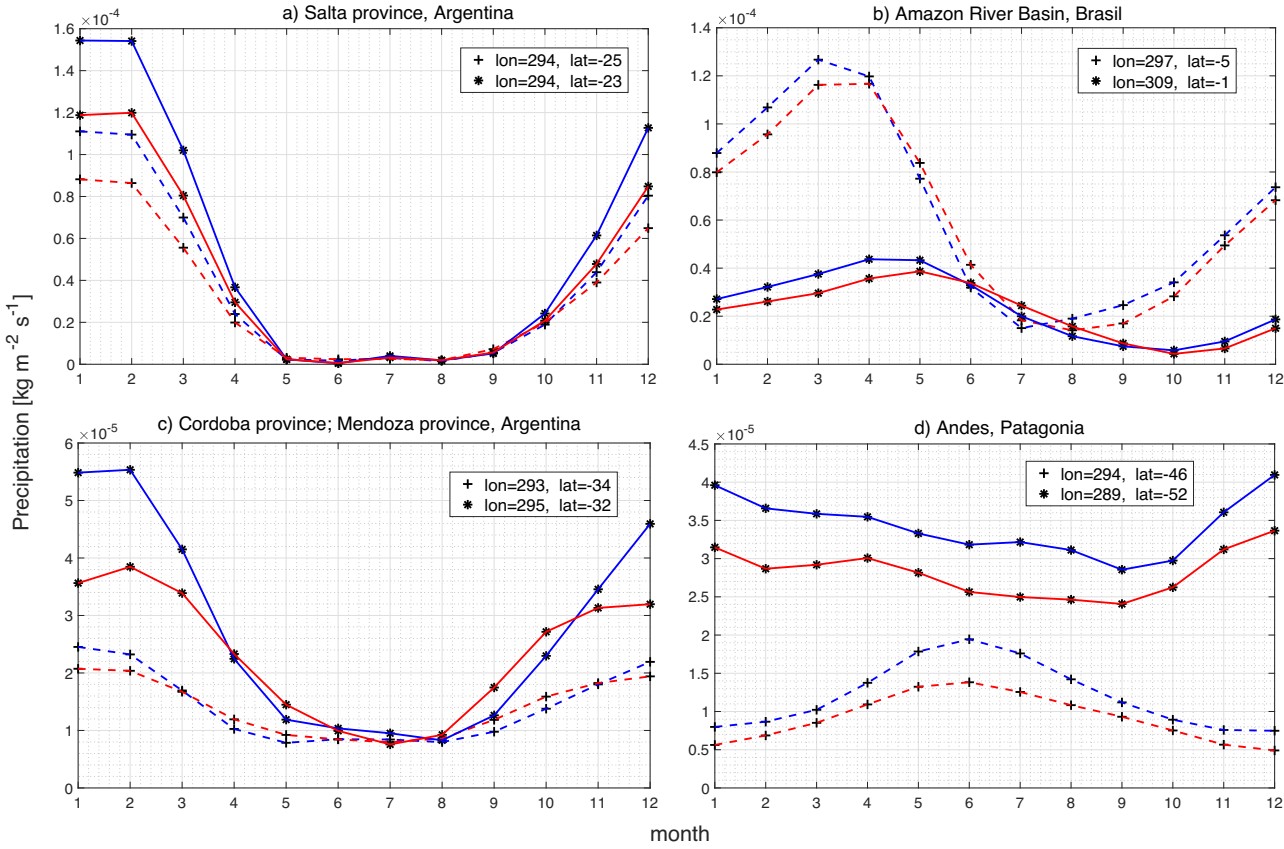

**Fig. 9 | Evolution of reconstructed precipitation over the years 1900 and 2010 for four regions in South America display distinctive change due to trends.** Each panel displays reconstructed precipitation time series for two locations within a region (plotted with a dashed line marked with " + "), or a solid line marked with "*"

as follows: (**a**) Salta province in northwest Argentina; (**b**) equatorial Amazon River basin; (**c**) Cordoba and Mendoza region, west Argentina; and (**d**) Patagonia. Blue and red lines depict precipitation reconstructed by the collection of periodic and trend eigenfunctions for year 1900 and 2010, respectively.

complex eigenfunction $v_j$ has a corresponding eigenfrequency $\omega_j \in \mathbb{R}$ and an eigenperiod $T_j = 2\pi/\omega_j$.

### Revealing climate change trends through eigenfunctions

Figure 7a, d displays the time series of eigenfunction $v_6$ of the generator along with globally averaged SST (Fig. 7a) and SAT (Fig. 7b) anomalies computed relative to a 1971–2000 monthly climatology. It is evident that $v_6$ represents a nonstationary pattern that correlates positively and significantly with the evolution of globally-averaged SST and SAT on decadal timescales. Notable features of the $v_6$ evolution include a modest amount of cooling over the first decade of the 20th century, followed by a warming episode from the mid-1930s to the late 1940s and a period of more sustained warming starting in the 1950s and lasting through the end of the analysis interval. This latter period is marked by episodes of rapid warming such as 1975–1980 and 2013–2018, but also contains intervals of warming slowdown, including the apparent warming "hiatus" that took place in the first decade of the 21st century. An analogous set of computations were made by estimating the transfer operator $\mathcal{L}$ of six-monthly evolution of SST anomaly fields. The second eigenvector (the leading nontrivial eigenvector) extracts the warming trend; see Supplementary Note 1 and Supplementary Fig. 2 for details. The features mentioned above are broadly consistent with accepted climate change trends over the industrial era[42–44], and demonstrate the capability of autonomous operator-theoretic techniques to provide nonparametric representations of nonstationary signals generated by non-autonomous dynamical systems.

Next, we consider spatial visualizations of the SST and SAT trends associated with $v_6$ obtained from the reconstruction procedure

described in Methods. This procedure is closely related to the reconstruction methods employed in EEOF analysis and SSA[51], and in essence involves projecting a target observable of interest to the cyclic subspace generated by a collection of one or more approximate Koopman/transfer operator eigenfunctions. In Fig. 7b, c we show snapshots of reconstructed global SST anomalies obtained from $v_6$ for January 1919 and 2019, respectively; Fig. 7e, f shows the corresponding patterns of global SAT anomalies. These patterns capture the climate warming that has taken place over industrial era, particularly over Arctic land masses (see Fig. 7f). Finer-grained features, such as atmospheric cooling in the Northeast Atlantic and Antarctic Peninsula are also clearly visible. Analogous industrial-era trend reconstructions to Fig. 7 were performed in earlier studies[65,66] using SSA with decadal embedding windows. In particular, ref. 65 raised the possibility of interdecadal oscillations, leading to warming hiatuses that hinder unequivocal detection of climate warming. Figure 3a of ref. 65 and Fig. 4a of ref. 66 depict SSA-derived trend time series that are generally more nuanced than the eigenfunction time series in Fig. 7a and Supplementary Fig. 2a (possibly due to the use of decadal vs. interannual embedding windows), but exhibit similar qualitative behavior in time intervals common to both studies.

### Response of seasonal cycle of precipitation to climate change

The seasonal cycle of the Earth's climate is driven by periodic variations of solar insolation between the northern and southern hemispheres over each year. The response of climatic variables to this driving depends on many factors and varies significantly between regions, even at the same latitude. In subtropical regions, monsoons are important examples of dynamical complexity[67], driven by land–sea

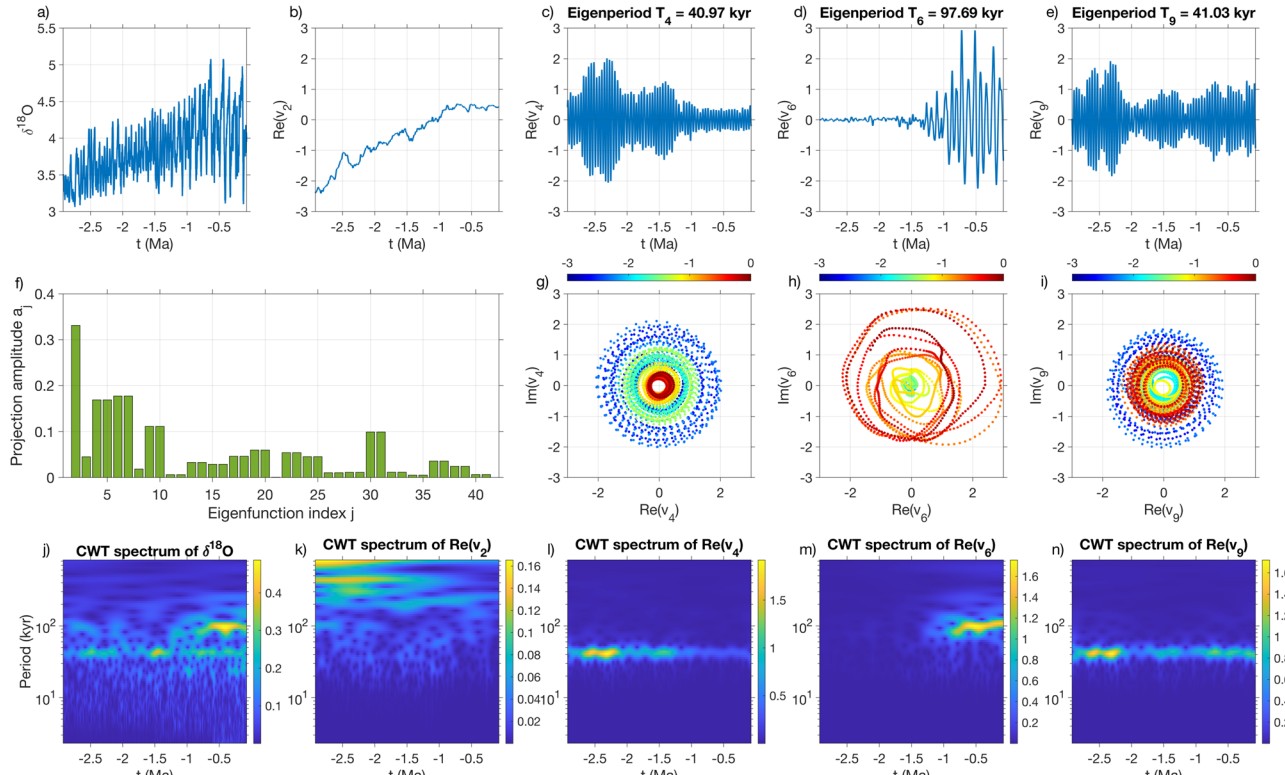

**Fig. 10 | Operator-theoretic analysis of $\delta^{18}O$ data from the LR04 stack over the past 3 Myr. a** Raw $\delta^{18}O$ time series over the past 4 Myr. **b** Time series of eigenfunction $v_2$ of the generator representing trend. **c**–**e** Time series of real parts of oscillatory eigenfunctions $v_4$, $v_6$, and $v_9$, respectively. **f** Moduli $a_j$ of the projection coefficients of the first 40 nonconstant eigenfunctions onto the $\delta^{18}O$ time series. **g**–**i** Two-dimensional phase space plots of $v_4$, $v_6$, and $v_9$ in the complex plane, respectively, colored by time. CWT spectra of the $\delta^{18}O$ time series (**j**) and the time series of eigenfunctions $v_2$, $v_4$, $v_6$, and $v_9$ (**k**–**n**).

contrasts and complex orography, and are the primary source of precipitation. Meanwhile, mid-latitude weather is dominated by seasonally modulated fronts and the jet stream, the properties of which depend strongly on meridional temperature gradients. As a result, even minor changes of the large-scale spatiotemporal structure of the seasonal cycle can lead to considerable impacts on a regional level, particularly with respect to changes in precipitation.

In this section, we use the eigenfunctions extracted from Indo-Pacific SST to characterize changes in seasonal precipitation in South America over the industrial era. As this region is strongly influenced by ENSO, this trend-driven signal can be possibly concealed in the raw data and hard to extract with traditional data analysis techniques. Here, our strategy is to project historical precipitation fields onto eigenfunctions associated with seasonality and inferred long-term trend in order to isolate the desired signal from ENSO and other nonperiodic modes of variability. In interpreting these results, the reader should keep in mind that, as with any precipitation reanalysis product, 20CR3v3 is subject to uncertainties and systematic biases. See ref. 68 for a comparison of precipitation fields from 20CRv3 and other popular reanalysis products.

Figures 8, 9 show reconstructions of precipitation fields over South America based on a collection of eigenfunctions that represent the seasonal climatology and the nonstationary trend—see the corresponding spectrum in Supplementary Fig. 1 and Supplementary Table 1. In more detail, in Fig. 8, we show 2D snapshots of reconstructed precipitation fields over South America based on (i) the constant eigenfunction $v_1$ representing the time-independent climatology; (ii) the complex-conjugate pair $\{v_2, v_3\}$ of annual periodic eigenfunctions (eigenperiod $T_j = 1$ yr) and their semiannual and triannual harmonics, $\{v_4, v_5\}$ and $\{v_9, v_{10}\}$ with eigenperiods $T_j = 1/2$ yr and $T_j = 1/3$ yr, respectively; and (iii) the trend eigenfunction $v_6$ from Fig. 7

and the complex-conjugate pairs $\{v_7, v_8\}$ and $\{v_{11}, v_{12}\}$ representing products between $v_6$ and the annual and semiannual pairs. These latter pairs behave analogously to a family of product modes between ENSO and the seasonal cycle called "combination modes"[69]. In the case of $\{v_7, v_8\}$ and $\{v_{11}, v_{12}\}$, the combination is between the *trend* represented by $v_6$ and the seasonal cycle, so it is natural to interpret them as "trend combination modes". These patterns explain a relatively small amount of variance of the raw Indo-Pacific SST data (about two orders of magnitude smaller than the seasonal cycle pair $\{v_1, v_2\}$) and, as we have verified in separate calculations, are not captured in the SSA/EEOF spectrum (possibly since standard SSA/EEOFs optimize for explained variance rather than slow correlation decay captured in the transfer/Koopman operator spectra; see the end of Supplementary Note 3 for further discussion). Nonetheless, as we explain below, the trend combination modes $\{v_7, v_8\}$ and $\{v_{11}, v_{12}\}$ play a major role in reconstructing the *change* of seasonal variability in response to the trend.

We interpret the reconstructions based on the union of eigenfunctions from (i), (ii), and (iii) as a time-dependent seasonal cycle associated with the trend represented by eigenfunction $v_6$. The snapshots in Fig. 8a, b are taken in January 1900 and January 2010, respectively; those in Fig. 8d, e are taken in July of the same years. Thus, differences between Fig. 8b, a, shown in Fig. 8c, provide a representation of how South American seasonal precipitation patterns in austral winter have changed over the past century, and differences between Fig. 8e, d, shown in Fig. 8f, represent the corresponding changes in austral summer. The latter, is the active period of the South American Monsoon System (SAMS). Note that the trend combination modes play a dominant role in the spatial patterns shown in Fig. 8c, f. This is because the contributions of the periodic eigenfunctions $v_2, v_3, v_4, v_5, v_9, v_{10}$ cancel out when computing the differenced fields from 1900 and 2010 at the same calendar month (January and July,

respectively). In Fig. 9, we show monthly time series of reconstructed precipitation fields sampled in representative locations in South America in 1900 (blue lines) and 2010 (red lines). For the remainder of this section, the terms summer and winter will refer to austral summer and winter, respectively.

The reconstructed spatial maps and time series show significant changes of the seasonal precipitation cycle in certain regions. First, as can be seen in the region 70°S–10°N and east of 70°W in Fig. 8c, the northeast parts of South America are characterized by increase of precipitation in the active SAMS season. In contrast, the Amazon river basin is drier during winter, as shown in Fig. 8f. Further south, desert regions east of the Andes, such as the Salta province shown in Fig. 9a, are characterized by absence of precipitation in winter and strong precipitation in summer−the latter, diminishes significantly between 1900 and 2010. Figure 9(c) shows precipitation time series at representative locations encompassing the wine regions of Cordoba and Mendoza province, where drying has been impacting the wine industry in recent years[70]. Our analysis also shows significant drying of the southern Andes and Patagonia (see Fig. 8), which is in agreement with the recent observational record[71]. Possible explanations for the changes in precipitation seen in Fig. 8 include intensification of the ascending branch of the Hadley Cell in the tropics[72], shifts in the spatial patterns of the Intratropical Convergence Zone (ITCZ) (and thus changes in SAMS), and changes of the Pacific South American (PSA) pattern[73–75] impacting the midlatitudes (in particular, the South Atlantic Convergence Zone).

## Quaternary glaciation cycles

The Quaternary geologic period, extending from ~2.6 million years ago (Ma) to the present, is characterized by the presence of glacial−interglacial cycles marked by the growth and decay of continental ice sheets in the northern hemisphere (NH). From the onset of the Quaternary to ~1 Ma, these cycles occurred with a fairly regular periodicity of approximately 41 thousand years (kyr). However, following a transition period known as the mid-Pleistocene transition (MPT), the NH glaciation cycles switched to a predominantly 100 kyr periodicity and became significantly more asymmetric and temporally irregular[76]. While details of the physical mechanisms underpinning the Quaternary glaciation cycles, including the MPT, remain elusive, they are generally thought to be the outcome of different types of orbital forcings of the Earth's climate system in conjunction with the prevalent atmospheric and geologic conditions such as $CO_2$ concentration and regolith distribution[77].

More specifically, the dominant orbital forcings of the Earth's climate include (i) orbital precession, with a main period of ~23 kyr; (ii) axial tilt (obliquity), with a main period of ~41 kyr; and (iii) orbital eccentricity, with a main period of ~100 kyr. The response of NH glacial sheets to these forcings over the Quaternary is thought to have been affected by two major factors, namely (i) reduced $CO_2$ concentration, possibly due to reduced outgassing from volcanoes; and (ii) reduction in NH regolith cover, possibly due to removal by erosion and/or glaciation. In particular, regolith makes ice sheets more mobile and thus more susceptible to orbital forcing. Recent studies[77] have posited that high $CO_2$ concentration and NH regolith thickness occurring in the early part of the Quaternary are more susceptible to a linear response to the 41 kyr axial tilt forcing, whereas lower $CO_2$ concentration and NH regolith thickness occurring in the post-MPT period resulted in higher susceptibility to 100 kyr orbital forcings with an associated nonlinear/asymmetric response of NH glacial sheets.

We analyze a scalar time series of $\delta^{18}O$ oxygen isotope ratio derived from the "LR04" stack of globally distributed benthic $\delta^{18}O$ records of Lisiecki and Raymo[31]. The LR04 dataset spans the past 5.3 Myr and is sampled non-uniformly in time with sampling intervals ranging from 1 kyr over the past 600 kyr to 2 kyr or longer further out in the past. Here, we analyze the past 3 Myr of the LR04 stack. For the

purposes of delay embedding, we interpolate the data to a fixed 1 kyr sampling interval using linear interpolation, leading to a $\delta^{18}O$ time series of 3001 samples for analysis depicted in Fig. 10a. It is clear that the time series exhibits oscillations with a drift in the mean and amplitude, as well as a frequency change around 1 Ma corresponding to the MPT. To examine these features in more detail, Fig. 10j displays a continuous wavelet transform (CWT) spectrum of the $\delta^{18}O$ time series where the ~40 kyr and ~100 kyr glaciation cycles are clearly evident. Note that while the 100 kyr cycle is predominantly active in the post-MPT portion of the time series (i.e., after ~1 Ma), the 40 kyr cycle exhibits activity throughout the analysis interval and there is coexistence of the two cycles in the post-MPT record. Our idealized model representing this behavior is Model F′ described in Methods and depicted in Fig. 1.

## Glaciation cycles from eigenfunctions

We compute approximate eigenfunctions $v_j$ of the generator by applying the same kernel-based approach as in the SST analysis to the $\delta^{18}O$ time series from the LR04 dataset, shown in Fig. 10a. Figure 10b−e shows time series of the real parts of representative generator eigenfunctions $v_j$. These eigenfunctions were chosen on the basis of (i) the amplitudes $a_j = |Y_j|$ of the projection coefficients $Y_j = \langle v'_j, h \rangle$ onto the observations $h$, where $v'_j \in \mathbb{C}^N$ is the dual (biorthonormal) eigenvector to $v_j$ (see "Methods"); and (ii) their frequency content in relation to known orbital-forcing frequencies. The amplitudes $a_j$ are displayed in Fig. 10f. In more detail, eigenfunction $v_2$ in Fig. 10b is a real eigenfunction (of zero corresponding eigenfrequency $\omega_2$) that has the largest projection amplitude among all non-constant eigenfunctions of the generator. The eigenfunctions in Fig. 10c−e are members of the complex-conjugate pairs that have the second to fourth largest projection amplitudes, respectively. Figure 10(g−i) shows trace plots of these eigenfunctions in the complex plane. Figure 10(k−n) shows CWT spectra of the eigenfunction time series in Fig. 10b−e, respectively.

It is clear from Fig. 10b that eigenvector $v_2$ represents a secular trend that is consistent with the trend in the mean of the $\delta^{18}O$ concentration signal. Meanwhile, eigenfunctions $v_4$ (Fig. 10c, g, l) and $v_6$ (Fig. 10d, h, m) feature narrowband oscillatory signals which are concentrated in the pre- and post-MPT periods and capture the 40 kyr and 100 kyr glaciation cycles, respectively. The corresponding eigenperiods, $T_4 \approx 41$ kyr and $T_6 \approx 98$ kyr, are also consistent with the dominant pre- and post-MPT periodicities.

We also carried out an analogous experiment using an approximation of the transfer operator $\mathcal{L}$ from an embedded $\delta^{18}O$ time series; see Supplementary Fig. 3. The leading eigenfunctions of the estimated $\mathcal{L}$ extract a secular trend shown in Supplementary Fig. 3b and distinct cycles – with eigenperiods of $T_4 \approx 99$ kyr and $T_6 \approx 41$ kyr – shown in Supplementary Figs. 3(c−d). Analogous trace plots of the complex eigenfunctions in the complex plane are illustrated in Supplementary Figs. 3(e−f). CWT spectra are given in Supplementary Figs. 3(g−j).

Besides $v_3$, $v_4$, and $v_6$ (and the complex conjugates of the latter two eigenfunctions, $v_5$ and $v_7$, respectively), there are other complex-conjugate pairs in the generator spectrum that project strongly onto the $\delta^{18}O$ time series. The first of these pairs, $\{v_9, v_{10}\}$ has an eigenperiod $T_9 \approx 41$ kyr which is close to the pre-MPT periodicity identified from $\{v_3, v_4\}$, but the amplitude of $\{v_9, v_{10}\}$ is more evenly distributed over the analysis interval, and in particular extends into the post-MPT period (see Fig. 10e, i, n). Further down in the spectrum of the generator there are eigenfunctions that oscillate at the ~23 kyr orbital precession periodicity, as well as eigenfunctions oscillating at intermediate timescales to the 40 kyr and 100 kyr cycles. These eigenfunctions are predominantly active in the post-MPT period, but they capture less variance than the eigenfunctions shown in Fig. 10 so we do not discuss them further here.

Overall, we find a qualitative difference in the pre- and post-MPT behavior of the dominant oscillatory eigenfunctions: The dominant

eigenfunctions exhibiting strong activity in the pre-MPT period, i.e., $v_4$, $v_5$, $v_9$, and $v_{10}$, have a narrowband frequency spectrum concentrated at approximately (40 kyr)$^{-1}$. On the other hand, the eigenfunctions $v_6$, $v_7$, $v_9$, and $v_{10}$ (as well as eigenfunctions further down the spectrum), which are active in the post-MPT period exhibit a significantly more diverse range of frequencies, approximately (100 kyr)$^{-1}$ to (30 kyr)$^{-1}$. This behavior is consistent with the higher irregularity of NH glaciation cycles known to occur since the MPT.

Established statistical techniques used to study Quaternary glaciation cycles (e.g. SSA[34], CWTs[77,78], moving Fourier spectral analysis[76], and Bayesian inference[79]) have been applied to data sources ranging from observational data such as the LR04 stack to output from conceptual models[80] and comprehensive paleo-climate models[77]. The nature of glaciation cycles has also been studied via a plethora of state-space dynamical systems techniques[78,81]. Among these statistical and dynamical methods, perhaps the most closely related to our Koopman/transfer operator approach is SSA. See Supplementary Note 3 for a comparison between SSA results applied to $\delta^{18}$O data (depicted in Supplementary Fig. 4) and the Koopman/transfer operator eigenfunctions in Fig. 10 and Supplementary Fig. 3.

In Supplementary Note 4 and Supplementary Fig. 5, we reconstruct the $\delta^{18}$O time series using the approach described in methods for various combinations of eigenfunctions from Fig. 10. These reconstructions demonstrate the efficacy of the eigenfunctions to recover important features of the $\delta^{18}$O evolution, including secular trend, frequency transition, and post-MPT irregularity.

## Discussion

Operator-theoretic techniques designed for analysis of data generated by autonomous dynamical systems have proven to be highly successful in diverse science and engineering applications. Using geometrical arguments, idealized models, and present and past climate data, we have shown that these methods can remain powerful analytical tools even when dealing with systems influenced by time-dependent exogenous factors. In particular, our computations are derived from a *single* observed time series, which is advantageous for the analysis of natural systems such as the Earth's climate system. In such systems there is only one observed history and it is challenging to well-sample ensembles of multiple likely driving conditions.

Central to our approach has been a combination of ideas from time-series delay-embedding geometry, Markov diffusion processes, spectral theory of transfer/Koopman operators, and kernel-based approaches for regularization of dynamical operators of autonomous deterministic systems. We use time delays to embed the observed time series in a higher-dimensional space where changes in the underlying dynamics carry distinct geometrical features (see Fig. 3 (right), Fig. 4 (right), and Fig. 5 (right)). By judiciously applying diffusive regularization to transfer and Koopman operators in the delay-embedding space, we compute eigenfunctions that separately encode nonstationary trends and long-lived cycles, and which are associated with the diffusion and drift components, respectively, of the regularized operators. In addition, spectral decomposition of the regularized operators yields product eigenfunctions (trend combination modes) that capture the modulation of the system's fundamental cycles by the nonstationary trends.

We illustrated the above ideas through idealized models that were chosen as surrogates of nonautonomous dynamics in two challenging real-world problems: Climate change occurring over the industrial era and the mid-Pleistocene transition (MPT) of Quaternary glaciation cycles. We demonstrated that eigenfunctions of transfer operators can successfully recover trends in systems undergoing drifts in the mean and amplitude of their oscillatory dynamics; (Models M and A, respectively). One of our main theoretical results (Lemma 1) shows that low-dimensional delay embedding

successfully delineates non-autonomous frequency switching in systems with oscillatory dynamics; (Model F; see Figs. 1, 6).

We then demonstrated the utility of our methods directly on Indo-Pacific SST and $\delta^{18}$O radioisotope concentration data. The Indo-Pacific SST analysis yielded eigenfunctions that provide a nonparametric representation of climate change trends. In addition, these eigenfunctions have associated families of product modes capturing the response of the seasonal cycle to the trend. Using these eigenfunction families, we reconstructed spatiotemporal patterns revealing significant regional changes in South American seasonal precipitation which may be explained by meridional shifts of the ITCZ and PSA pattern. Meanwhile, the $\delta^{18}$O analysis yielded eigenfunctions representing the long-term trend of benthic $\delta^{18}$O over the past 3 million years, as well as the fundamental 40 kyr and 100 kyr glaciation cycles occurring before and after the MPT, respectively. One of the key findings of this analysis was the presence of multiple coexisting cycles in the post-MPT period ( ~ 1 Ma to the present), which could help explain the irregularity of glaciation cycles in that period.

In conclusion, the theory and results described in this paper demonstrate the ability of autonomous operator-theoretic techniques to extract trends and cycles from certain classes of non-autonomous systems. Importantly, these autonomous methods use information from a single time series, enabling them to be applied in many natural science domains such as climate dynamics where repeated experiments are impossible.

## Methods
### Model F′: switching between two coexisting oscillation frequencies
Model F described dynamics switching between two rotation frequencies on two halves of a cylinder, where the phase around the cylinder was read off by a fixed observation function. One could alternatively move the nonstationarity out of the dynamics and into the observation function; we call this alternative "Model F′". The dynamics $T : [0, 1] \times S^1 \times S^1 \to [0, 1] \times S^1 \times S^1$ of Model F′ occur on an enlarged phase space – a solid 2-torus $[0, 1] \times S^1 \times S^1$, see Fig. 1 (center right) –to allow for two frequencies to always exist, while the observation function $h : [0,1] \times S^1 \times S^1 \to \mathbb{R}$ predominantly records one frequency or the other. Explicitly, we have

$$T(x, \theta) = (f(x), \theta_1 + \alpha_1, \theta_2 + \alpha_2),\qquad(6)$$

and

$$h(x, \theta_1, \theta_2) = w(x) \cos(2\pi\theta_1) + (1 - w(x)) \cos(2\pi\theta_2).\qquad(7)$$

**Proof of Lemma 1.** For $0 < \beta \le \pi/2$ the elliptical images $e_\beta \subset \mathbb{R}^3$ of a unit circle under $\Phi_\beta$ lie in the plane $p_\beta$ spanned by $[1, \cos(\beta), \cos(2\beta)]$ and $[0, -\sin(\beta), -\sin(2\beta)]$. As $\beta$ varies, this family of planes always contain the fixed vector $[1, 0, -1]$. One may compute that the vector $[\sin(2\beta), 2\sin(\beta), \sin(2\beta)]$ is orthogonal to $[1, 0, -1]$, and that together they span $p_\beta$. Note that the vectors $[\sin(2\beta), 2\sin(\beta), \sin(2\beta)]$ are distinct for distinct $0 < \beta \le \pi/2$. This means that for $\beta \ne \beta' \in (0, \pi/2]$ the only intersection of $p_\beta$ and $p_{\beta'}$ (and therefore the only possible intersection of $e_\beta$ and $e_{\beta'}$) is along the vector $[1, 0, -1]$.

The behavior of the $e_\beta$ is as follows. For $\beta = 0$, one obtains a degenerate ellipse (a line segment) passing through the origin in the direction $[1, 1, 1]$ (this is the $\beta \to 0$ limiting direction of $[\sin(2\beta), 2\sin(\beta), \sin(2\beta)]$). For $0 < \beta \le \pi/2$ one obtains an ellipse with one axis in the direction $[\sin(2\beta), 2\sin(\beta), \sin(2\beta)]$ of length $\sqrt{2 + \cos(2\beta)}$ and the other axis in the direction $[1, 0, -1]$ of length $\sqrt{1 - \cos(2\beta)}$. These two lengths are the singular values of the transformation $\Phi_\beta$. As the planes $p_\beta$ (and the ellipses $e_\beta$ contained in them) pivot about the fixed vector $[1, 0, -1]$, the length of their axis in this

direction increases monotonically from $\beta = 0$ to $\beta = \pi/2$. This axis length increase combined with the pivoting about a common axis direction implies that they are disjoint for $0 < \beta \neq \beta' \leq \pi/2$.

One can verify that the product of the above two singular values of $\Phi_\beta$ has a unique maximum in the range $\beta \in [0, \pi/2]$ at $\beta = \pi/3$, where this product has the value 1.5. The angle $\beta = \pi/3$ is also the unique angle in $[0, \pi/2]$ where the singular values coincide, and therefore also the unique angle for which the ellipse $e_\beta$ is a circle. □

## Reconstruction of observables by projection onto eigenfunctions

Let $v_i \in \mathbb{C}^N$ denote an eigenvector onto which we wish to project, and let $v'_i \in \mathbb{C}^N$ denote the corresponding eigenvector of the adjoint transfer operator $P^\top$ (or the adjoint generator), where we normalize so that $v'^\dagger_i v_i = 1$. Here, $\dagger$ is the complex-conjugate transpose. Because $v'^\dagger_i v_j = 0$ for $i \neq j$, the standard rank-1 projection matrix $\Pi_i := v_i v'^\dagger_i$ has range span$\{v_i\}$ and nullspace span$\{v_j\}_{j \neq i}$. We may now project an arbitrary *scalar-valued* time series $\{y_n\}_{n=1}^N$ represented by the $N$-vector $y$ onto $v_i$ by acting on $y$ with matrix multiplication by $\Pi_i$ according to $(\Pi_i y)_m = \sum_{n=1}^N \Pi_{mn} y_n = v_{i,m} \cdot (\sum_{n=1}^N \overline{v'_{i,n}} y_n)$. To project *vector-valued* time series (e.g., SST images) $\{Y_n\}_{n=1}^N$ where each $Y_n \in \mathbb{R}^d$, we act on $Y \in \mathbb{R}^{N \times d}$ in an analogous way, applying the projection $\Pi_i$ componentwise in $\mathbb{R}^d$: $(\Pi_n(Y))_m := v_{i,m} \cdot (\sum_{n=1}^N \overline{v'_{n,i}} Y_n) \in \mathbb{C}^d$. In addition, given a collection $\{v_{i_1}, \ldots, v_{i_r}\}$ of distinct eigenvectors, we may project onto the $r$-dimensional subspace span$\{v_{i_1}, \ldots, v_{i_r}\}$ by forming the rank-$r$ projection matrix $\Pi = \sum_{j=1}^r \Pi_{i_j}$ and acting with it on $Y$ componentwise: $(\Pi(Y))_m = \sum_{j=1}^r (\Pi_{i_j}(Y))_m = \sum_{j=1}^r v_{i_j,m} \cdot (\sum_{n=1}^N \overline{v'_{i_j,n}} Y_n)$. If the collection $\{v_{i_1}, \ldots, v_{i_r}\}$ consists of real eigenvectors and/or complex-conjugate pairs, then $\Pi(Y)$ is real. If we employ Takens embedding on the time series $\{Y_n\}$ in order to improve state estimation, the above action of $\Pi$ is again applied componentwise to the delay vectors formed from concatenations of various $Y_n$. See Methods in ref. 25, or ref. 51, for further details.

## Indo-pacific SST calculations

We analyze a time series $h_1, \ldots, h_{\tilde{N}}$ of monthly-averaged Indo-Pacific SST, where $h_1$ is the snapshot for January 1891 and $h_{\tilde{N}}$ with $\tilde{N} = 1548$ is the snapshot for December 2019. Each analyzed SST snapshot $h_t$ contains $d = 4868$ grid points, i.e., it is considered as a point in $\mathbb{R}^{4868}$. Using this data, we compute data-driven approximations of the transfer operator and the generator of the Koopman/transfer operator groups using the methods described in ref. 25. Results from the generator calculations are discussed in the main text. The transfer operator results are broadly consistent with those from the generator and are described in the Supplementary Information. A full listing of the parameters used in our experiments is included in Supplementary Table 1.

**Generator approximation.** For computation of the eigenvectors of the generator we apply Takens delay embedding to the snapshots with $Q - 1 = 47$ lags of duration $\ell = 1$ month. We thus obtain an embedding window length of 4 yr and corresponding SST "videos" of dimension $dQ = 233,664$. This choice of embedding parameters was previously found[25] to provide adequate representation of annual and interannual processes such as the seasonal cycle and ENSO, respectively, as well as longer-term processes such as climate change trends and Pacific decadal variability. We use the delay-embedded data to build a kernel matrix of size $N \times N$, $N = \tilde{N} - Q + 1 = 1501$, whose top $L = 401$ eigenvectors provide a data-driven basis of observables for representing the generator as an $L \times L$ matrix with respect to this basis. The result is a collection of eigenvectors $v_1, v_2, \ldots, v_L \in \mathbb{C}^N$ and corresponding eigenvalues $\gamma_1, \gamma_2, \ldots, \gamma_L \in \mathbb{C}$, where $-\text{Re}\gamma_i$ represents decay rate and

$\text{Im}\gamma_i$ represents oscillatory frequency. The spectral mapping theorem connects the rates represented by the eigenvalues $\gamma_i$ of the generator with the eigenvalues of the time-$t$ transfer operator $\mathcal{L}^t$ through $\lambda_i = e^{-i\gamma_i t}$. In our computations, we order the eigenpairs $(\gamma_i, v_i)$ in order of increasing decay rate (i.e., decreasing $\text{Re}\gamma_i$). We note that our numerical results are not particularly sensitive on the choice of $Q$ and $L$, and qualitatively similar results can be obtained for $Q$ in the range 24–120 and $L$ in the range 200–500.

**Transfer operator approximation.** For the construction of the transition matrix $P$ we use a single lag, setting $Q = 2$ to minimize the sparsity of the data, producing embedded data points in $\mathbb{R}^{9736}$. We use a lag duration of $\ell = 12$ months, which is about one quarter of the average period of the ENSO cycle. To reduce the effect of noise, we step forward $s = 6$ months, and we slightly reduce the neighborhood size to $K = 5$ to obtain more accurate estimates of the ENSO cycle. Supplementary Fig. 2 displays the leading nontrivial real eigenvector (corresponding to eigenvalue $\lambda_2 = 0.4135$). This second eigenvalue is relatively far from unity because of the sparsity of the high-dimensional data; nevertheless the corresponding eigenvector is consistent with a qualitative description of the warming trend over the last 150 years, as illustrated in Supplementary Fig. 2 through comparisons with global SST and global SAT.

## Benthic $\delta^{18}$O calculations

The time series of $\delta^{18}$O concentration derived by interpolation of the LR04 stack is scalar-valued ($d = 1$), and consists of $\bar{N} = 3001$ samples taken every 1 kyr over the last 3 Myr. We compute approximate eigenfunctions of the transfer operator and the generator using the same methods as in the Indo-Pacific SST experiments. A full listing of the parameters is provided in Supplementary Table 3.

**Generator approximation.** We use a delay embedding window spanning 150 kyr (i.e., $Q = 150$ and $\ell = 1$ kyr). This embedding window was chosen on the basis of being comparable to the ~100 kyr characteristic timescale of the post-MPT glaciation cycles, though our results are not too sensitive on this choice. After embedding, the number of samples available for analysis is $N = \tilde{N} - Q + 1 = 2851$. We use the leading $L = 81$ eigenvectors of the associated $N \times N$ kernel matrix to approximate the generator as in the Indo-Pacific SST analysis.

**Transfer operator approximation.** We embed the relatively noisy scalar signal $h_t$ in five dimensions so as to reduce false neighbors. We use a fixed lag $\ell = 10$ kyr (10 sampling intervals) chosen approximately according to the considerations following Lemma 1. To further reduce the effect of noise we step forward 7000 years at a time ($s = 7$ sampling intervals) when constructing the transition matrix $P$; $K = 7$ nearest neighbors are used to set the variable bandwidth. The eigenfunction corresponding to the second eigenvalue extracts the $\delta^{18}$O trend in the mean of the signal; see Supplementary Fig. 3 for a comparison between the original signal and the mean trend. The two leading complex eigenvector pairs extract and separate the two dominant frequency regimes in the $\delta^{18}$O signal; see Supplementary Note 2 and Supplementary Fig. 3. The corresponding complex eigenvalues estimate cycle lengths of 98,640 and 40,775 yr, which match well with the accepted periods either side of the MPT. Thus the eigenvectors of the transfer operator have separated the trend in the mean of the $\delta^{18}$O time series from the oscillatory components. Further, the complex eigenvectors have separated and identified the two dominant frequencies, including indicating when each frequency is present through the amplitude of the eigenvectors.

## Data availability

The ERSSTv4, NCEP, and 20CRv3 reanalysis data are available at the National Centers for Environmental Information repositories, under

accession codes https://www.ncdc.noaa.gov/data-access/marineocean-data/extended-reconstructed-sea-surface-temperature-ersst-v4 and https://psl.noaa.gov/data/gridded/data.ncep.reanalysis.html, https://www.psl.noaa.gov/data/gridded/data.20thC_ReanV3.html, respectively. The $\delta^{18}O$ concentration data from the LR04 stack are available at the URLs https://lorraine-lisiecki.com/stack.html. The processed data can be generated by running the code in the repositories listed in the Code Availability statement. Processed data is also available from the corresponding author on request.

## Code availability
MATLAB code implementing transfer operator calculations for Models M, A, and F, as well as the ENSO and ice-core data is available at https://github.com/gfroyland/Trends. MATLAB code implementing the numerical approximation of the generator employed in the paper is available at https://dg227.github.io/NLSA/.

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

## Acknowledgements

G.F.'s research is partially supported by Australian Research Council Discovery Projects DP180101223 and DP210100357, an Einstein Foundation Visiting Fellowship, and the School of Mathematics and Statistics at UNSW Sydney. G.F. gratefully acknowledges generous support from the Jack Byrne Academic Cluster in Mathematics and Decision Science during a visit to Dartmouth College, and for kind local hospitality from the Department of Mathematics. D.G. acknowledges support from the U.S. National Science Foundation under grants 1842538 and DMS-1854383, the U.S. Office of Naval Research under MURI grant N00014-19-1-242, and the US Department of Defense, Basic Research Office under Vannevar Bush Faculty Fellowship grant N00014-21-1-2946. E.L. was supported as an undergraduate research assistant at New York University under the first and third of these grants. The authors thank Benjamin Lintner for stimulating discussions on the climate dynamics applications presented in the paper.

## Author contributions

G.F. and D.G. designed and performed the theoretical study and numerical experiments on idealized non-autonomous models. G.F., D.G. and J.S. designed the study on industrial-era climate dynamics,

performed the associated operator computations and interpreted the results. D.G. and J.S. designed the study on glaciation cycles. E.L. prepared the $\delta^{18}O$ dataset. G.F., D.G. and E.L. performed the associated operator computations and interpreted the results. G.F. and D.G. wrote the paper. All authors read and approved the final manuscript.

## Competing interests

The authors declare no competing interests.
