## [Peer Review File · Nature Communications]

REVIEWER COMMENTS

Reviewer #1 (Remarks to the Author):

General. This is a very interesting and well written paper. It represents a continuation of valuable work by the authors that introduced operator-theoretic methods to autonomous dynamical systems and to climate dynamics. The novelty in this paper is establishing the usefulness of such methods, with minor modifications, to nonautonomous dynamical systems, i.e., an extension from systems with no explicit time dependence in their forcing or coefficients to systems in which such explicit time dependence is present.

The paper illustrates the abstract situation via three idealized models for: (i) oscillation with a drift in the mean (Model M); (ii) oscillation with a drift in the amplitude (Model A); and (iii) switching between two distinct oscillation frequencies (Model F). These three models correspond to features associated with the two climate problems that are the target of the extended methods' application: (1) changes in the behavior of intrinsic climate variability due to anthropogenic effects during the industrial era; and (2) changes in the frequency and amplitude of Quaternary glaciation cycles during the Mid-Pleistocene (MPT) transition.

The climate data for these two problems are essentially projected onto a subspace spanned by a subset of the eigenvectors of the transfer and Koopman operators, L and K , associated with the pushforward and pullback of a class of functions $\{f\}$. The methodology is very similar with that of singular spectrum analysis (SSA) and multichannel SSA (M-SSA), as the authors do state. In the latter case, the L and K operators are replaced with the covariance operator of Karhunen-Loève (coincidentally K-L) theory. Among the paper's results, a truly novel one is that the characterization of seasonal changes in the industrial-era precipitation field over South America is achieved using only a subspace of eigenfunctions extracted from Indo-Pacific sea surface temperatures (SSTs).

Major Comments

1. Overall, the paper is quite clear and enjoyable to read. My major issue with it is that the two climatic problems it addresses are presented in a very incomplete and, especially for the MPT one, biased way. To start with, the framework of nonautonomous dynamical systems, both deterministic and stochastic, has been proposed about two decades ago by the research groups of Michael Ghil and Tamás Tél (Ghil et al., *Physica D*, 2008; Chekroun et al., *Physica D*, 2011; Bódai & Tél, *Chaos*, 2012; Tél et al., *JSP*, 2020) and there is by now a huge literature that has followed up. Some mention of this literature would be welcome.

2. While SSA is mentioned in passing, the connections merit further comparison. For instance, the issue of internal climate variability vs. anthropogenically forced climate change was treated via SSA by Ghil & Vautard (*Nature*, 1991), who showed that an episode of “warming hiatus” was quite likely well before its actual occurrence. More generally, what is the prognostic, as opposed to just diagnostic, power of the proposed operator-theoretic methods? SSA has been used, for instance, quite successfully in ENSO prediction (Barnston et al., *BAMS*, 1994; Jiang & Ghil, *GRL*, 1998).

3. Concerning the MPT, the paper only mentions the role of CO₂ and regoliths in its occurrence and proposes its own minimal model F as the basis of the analysis. Riechers et al. (*Clim. Past*, 2022) have reviewed in extenso the issue of intrinsic climate oscillations and their role in the MPT, in the context of nonautonomous dynamics. Their paper’s Table A1 extends over 3 pages and tenths of models, both with and without intrinsic oscillations. Can this paper’s methods help resolve, for instance, the issue of whether the MPT has been caused by sub- or supercritical Hopf bifurcation?

4. SSA is based on K-L theory, which requires at least weak stationarity of a random process to apply. Eckmann & Ruelle (*RMP*, 1985) have told us a long time ago that spectral theory can apply to deterministically chaotic dynamics, too. But SSA has been applied quite successfully to nonstationary time series, whether deterministically or stochastically generated; see the Ghil & Vautard (1991) results above as one of very many such applications.

In fact, SSA typically produces a nonlinear trend as the leading component of the decomposition in such cases; here, nonlinear means that the corresponding $x_1 = x_1(t)$ graph is not a straight line. It might be quite interesting to understand these “facts of life” better by using a novel and ingenious analogy or actual mathematical relationship between “Koopmania” and SSA. Santos Gutiérrez et al. (*Chaos*, 2021) have, in this spirit, helped understand the remarkable success — both diagnostic and prognostic — of empirical model reduction, originally a purely data-driven method, in the light of analytical, operator-theoretical methods.

Discussion and recommendation. The list of Major Comments above is clearly an ambitious one, justified by the quality of the paper at hand, in its current version. I think that the paper should certainly be published in a Nature Publishing Group journal. Which one should be determined by the improvements that the authors will choose to address in this paper, as opposed to some future ones. With this recommendation in mind, I’ve added a small selection of Minor Comments as an Appendix to this review.

Appendix. A few minor comments

p. 2, 1st full para. Benthic $\delta^{18}\text{O}$ radioisotope concentration is a proxy for global temperature and ice volume (e.g., Lisiecki & Raymo, Paleo, 2005, p. 2, para. [5]), not of atmospheric CO_2 concentration!

p. 4, 1st full para. The special circular arrow notation for a map T from Ω to itself is sexy, & I've happily used it myself at the blackboard. But it is replaced later in the paper with the more common $T: \Omega \rightarrow \Omega$ notation. Maybe use only the latter one?

p. 8, 1st l. after Eq. (4): Explain a bit "row-stochasticize"; maybe add an appendix or two to explain the more technical aspects of the paper to a wider climate-interested audience.

p. 14, Fig. 8. The panel labeling in the figure caption is not consistent with the text and the panel legends.

p. 15, Fig. 9. This figure & Fig. 8 are the most striking, & only truly novel, results of the application of the methods to the two climate problems. It might be worthwhile including in the possible explanations a weakening of the Pacific South American (PSA) anomaly pattern (Mo & Ghil, JAS, 1987; Li & Le Treut, GRL, 1999).

p. 17, Fig. 10. Panels (b)–(e) look amazingly like SSA results; see, for instance, Fig. 2 in Kondrashov et al. (GRL, 2005) for an oscillatory component or Fig. 4 of Moron et al. (Clim. Dyn., 1998) for industrial-era trends.

N.B. In SSA, the eigenvector associated with a trend has no sign change but is not identically equal to 1 (different version of the Perron-Frobenius theorem or some version of Sturm-Liouville theory); it is a data-adaptive set of positive weights that is likely to yield a better running mean than the operator-theoretic one. A closer look might be worth it.

Reviewer #3 (Remarks to the Author):

In their new manuscript, Froyland et al. present theoretical foundations and climate applications of a new operator-theoretic framework for decomposing time series into trends and persistent cycles, which radically differs from most established statistical time series decomposition techniques. There is a great level of novelty in this interesting work, and I may foresee vast potentials for further applications across various fields of science.

The paper is essentially well written, although I tend to struggle a little bit with the distinction between results and methods sections as requested by the journal's native manuscript format (for example, one may argue if all of the section on time-delay embedding fits into the results section). Also the order of figures and their referencing across the text does not always appear fully logical to me. However, I still think the authors did a good job given the corresponding requirements.

The few concerns and suggestions for improvements that I do have (listed below) amount to a recommendation for acceptance of this interesting work after minor revision.

General comments:

1. The operator based techniques developed and employed in this work are related to “eigenfunctions of Koopman and transfer operators” (abstract, l.12), but it is not quite clearly worked out that it is actually the transfer operator that is exploited in this work (if I understand this point correctly). I suggest revising the text accordingly, or alternatively clarify briefly the link between the eigenfunctions of both operators.
2. The concept of a trend is a somewhat ambiguous thing in statistics. My understanding is that it describes a specific component embedded in a time series that behaves completely monotonically over the considered observation period. While this may merely sound a subtlety, it may be relevant in interpretations of climate applications, where a wide audience may associate “trend” with anthropogenic effects beyond natural variability. While the presented technique seems powerful to identify such monotonic components in climate time series, any attribution of the so-called trend component to actual trends (long-term changes beyond the period of observations) versus, for example, multi-decadal variability may be problematic. My suggestion would be clarifying the concept of a “trend” in the context of the present work being a statistical one, not necessarily implying a trend in the sense of the common use of this term in climate sciences.
3. The terminology used with respect to the paleoclimate example is incorrect. $\delta^{18}\text{O}$ is not a radioisotope concentration, but the ratio between the respective concentrations of two oxygen isotopes in the studied material (O-18 over O-16). The benthic $\delta^{18}\text{O}$ stacked record studied here should also not be interpreted as a proxy of atmospheric CO₂ concentrations, but rather as representative of ambient water temperatures (while also being affected by water salinity and global ice volume). There exists a potential indirect dependency on atmospheric CO₂ concentrations, which is however non-trivial.
4. Regarding the paleoclimate application, it might be interesting to have a brief discussion on how the newly proposed technique behaves in comparison with other established statistical techniques like continuous wavelet transform, singular spectrum/system analysis, or empirical mode decomposition. I do not expect any additional analyses on this aspect, but would welcome some general perspectives on this from the authors, potentially along with references to published works (e.g. Nyman & Ditlevsen, *Climate Dynamics*, 2019; Mukhin et al., *Scientific Reports*, 2019). Some reference may be useful to the statement on the last sentence before the discussion (top of page 18: “...known to occur”).

5. On the interpretation of the eigenfunction families for the Indo-Pacific SST data: The authors associate their findings to “meridional shifts of the South American monsoon”. I am wondering if it is indeed the monsoon or rather the inter-tropical convergence zone that is shifting here. It may be worth looking a bit more into the seasonality of the underlying patterns. I would also be wondering if the proposed explanation of Fig. 8 in terms of Hadley cell and SACZ intensifications (bottom of page 15) can be supported by any reference to existing climatological studies.

Technical comments:

- It is not completely obvious to me what “slowly decaying variables” should be (already in the abstract, l.9) – maybe two or three additional words could clarify this
- Abstract: I suggest briefly mentioning the type of data (reanalysis, paleoclimate proxies) used in the two real-world case studies
- Page 5, l.2: “The same dynamical model T as in (1)...”
- Figure 6, caption: is it really two complex eigenvectors or two pairs of eigenvectors with complex-conjugate eigenvalues? (similar shortages may be found in other places of the text)
- Page 14: I think “austral” is commonly not capitalized in climate studies when referring to “southern hemispheric”
- Page 15, bottom: when referring to a point in deep time, one might commonly use, e.g., “1 Myr BP (before present)”, as opposed to references to time spans (“spanning ~2.6 million years (Myr)” – here “ago” should be removed)
- Page 20, Generator approximation: Should $\text{Im } \omega_i$ rather be $\text{IM } \gamma_i$ (which might be what the authors refer to as ω_i)?
- Page 23, ref. 25: check author
- Page 23, ref. 26: typo in “feedback”
- Suppl. Fig. 1: I think here is an inconsistent notation being used as compared to the main text (eigenvalues denoted by λ_i instead of γ_i)

Referee’s report on the paper “Revealing trends and persistent cycles of non-autonomous systems with operator-theoretic techniques: Applications to past and present climate dynamics” by Froyland et al.

The authors provide an application of Koopman and transfer operator frameworks for autonomous systems to analysis of non-autonomous systems. They then apply the framework to two real-world examples from present and past climate dynamics: Variability of sea surface temperature (SST) over the industrial era and the mid-Pleistocene transition (MPT) of Quaternary glaciation cycles. Separating forcing and internal variability aspects is, of course, a valuable endeavor. But it is not clear to me how much is new here on the methodological side, because the paper is quite sparse on referencing previous work by others on analysis of nonautonomous systems using Koopman operators. Here are some examples and additional comments

1. Page 1. There are a number of missing references on analysis non-autonomous systems using operator theoretic techniques: by typing e.g. "koopman and nonautonomous" into Google Scholar , we find Mezic and Surana (2016) is perhaps the first one, but many followed. What should be reviewed in addition is work on system identification for controlled systems (which are nonautonomous by defonition), see eg Mauroy et al book "Koopman operator in systems and control".
2. Page 2, “We will show in this present work that an estimate of the (non-stationary) Indo-Pacific warming trend, as well as the modulation of the seasonal cycle by the trend, can be obtained through eigenvectors of transfer or Koopman operators.”. Isn’t it clear how to do this since Mezic-Surana? Namely, a linear trend can be captured by adding a variable L describing that trend with dynamics $\dot{L} = 0$. Any cyclic forcings can also be captured that way, by using the notion of “suspension”. The authors use time-delays here, but that is not novel either - see the comment below on Hankel-DMD.
3. Pg 3, Figure 1, this kind of dynamics, and the application of autonomous Koopman operator methods to the nonautonomous fluid flow around a cylinder, can be found in Glaz et al (2017) paper “Quasi-periodic intermittency in oscillating cylinder flow”
4. Pg 5 “A classical approach to such a time series might be to embed it in a higher-dimensional space using Takens method of delays” . The paper does not point out a number of relevant references: there is a history of using time delays in Koopman operator context starting with Mezic and Banaszuk (2004), and extensions to EDMD were done in Arbabi and Mezic (2017) - the later under the name of Hankel-DMD, which is in massive use these days. Is there any difference with the methods developed here to approximate operators?

In summary, I do think this is an interesting topic, but it is unclear what aspects of methodology are actually new. I am open to reconsidering upon author's explanation.

Revealing trends and persistent cycles of non-autonomous systems with operator-theoretic techniques: Applications to past and present climate dynamics

Gary Froyland, Dimitrios Giannakis, Edoardo Luna, Joanna Slawinska

December 11, 2023

We thank the Reviewers and Editors for their careful consideration and comments on the manuscript. Below is a point-by-point response to the Reviewers' comments, using an italic font to indicate text copied verbatim from their reports. Changed or added text in the revised manuscript is highlighted in blue.

1 Response to Reviewer #1

General. This is a very interesting and well written paper. It represents a continuation of valuable work by the authors that introduced operator-theoretic methods to autonomous dynamical systems and to climate dynamics. The novelty in this paper is establishing the usefulness of such methods, with minor modifications, to nonautonomous dynamical systems, i.e., an extension from systems with no explicit time dependence in their forcing or coefficients to systems in which such explicit time dependence is present. The paper illustrates the abstract situation via three idealized models for: (i) oscillation with a drift in the mean (Model M); (ii) oscillation with a drift in the amplitude (Model A); and (iii) switching between two distinct oscillation frequencies (Model F). These three models correspond to features associated with the two climate problems that are the target of the extended methods' application: (1) changes in the behavior of intrinsic climate variability due to anthropogenic effects during the industrial era; and (2) changes in the frequency and amplitude of Quaternary glaciation cycles during the Mid-Pleistocene (MPT) transition. The climate data for these two problems are essentially projected onto a subspace spanned by a subset of the eigenvectors of the transfer and Koopman operators, L and K , associated with the pushforward and pullback of a class of functions $\{f\}$. The methodology is very similar with that of singular spectrum analysis (SSA) and multichannel SSA (M-SSA), as the authors do state. In the latter case, the L and K operators are replaced with the covariance operator of Karhunen-Loève (coincidentally K - L) theory. Among the paper's results, a truly novel one is that the characterization of seasonal changes in the industrial-era precipitation field over South America is achieved using only a subspace of eigenfunctions extracted from Indo-Pacific sea surface temperatures (SSTs).

We thank the Reviewer for their positive remarks on our work.

*1. Overall, the paper is quite clear and enjoyable to read. My major issue with it is that the two climatic problems it addresses are presented in a very incomplete and, especially for the MPT one, biased way. To start with, the framework of nonautonomous dynamical systems, both deterministic and stochastic, has been proposed about two decades ago by the research groups of Michael Ghil and Tamás Tél (Ghil et al., *Physica D*, 2008; Chekroun et al., *Physica D*, 2011; Bódai & Tél, *Chaos*, 2012; Tél et al., *JSP*, 2020) and there is by now a huge literature that has followed up. Some mention of this literature would be welcome.*

Response: The main purpose of this work is to explain why naive application of autonomous transfer and Koopman operator methods to nonstationary time series data can produce useful trend or switching information through their dominant eigenvectors, when there is no good *a priori* reason for this. Our illustrations are with two sets of climate data because this is one area where multiple trajectories are difficult to come by, but the methods and explanations of the submission go well beyond climate. Because of this focus it was not possible to include a large literature on nonautonomous systems in general or nonautonomous

climate models in particular. We have added a few more nonautonomous climate references on p. 12, going back to the book by Dymnikov and Filatov (1997), which is a little closer to our focus, as it discusses operator-theoretic techniques. The papers mentioned by the Reviewer are state-space (not function-space) methods and so are farther from the techniques and explanations we are introducing in this submission. In this revision, we have included references to Ghil et al. (2008) and Tél et al. (2020) to highlight state-space techniques that employ pullback attractors (see p. 12).

2. *While SSA is mentioned in passing, the connections merit further comparison. For instance, the issue of internal climate variability vs. anthropogenically forced climate change was treated via SSA by Ghil & Vautard (Nature, 1991), who showed that an episode of “warming hiatus” was quite likely well before its actual occurrence. More generally, what is the prognostic, as opposed to just diagnostic, power of the proposed operator-theoretic methods? SSA has been used, for instance, quite successfully in ENSO prediction (Barnston et al., BAMS, 1994; Jiang & Ghil, GRL, 1998).*

Response: We agree that the connections with SSA merit further discussion and we have added text to that effect in a number of instances. First, we introduce SSA and the related extended empirical orthogonal function (EEOF) analysis on p. 12. There, we also discuss connections between SSA and Koopman/transfer operators (which we elaborate on in our response to comment 4 below). On pp. 13–14, we cite and discuss the work of Ghil & Vautard (1991) in the context of warming hiatuses. With regards to predictive capabilities of operator-theoretic methods, we have added a discussion and references on p. 12. The cited references include some of our previous work on ENSO, MJO, and sea ice prediction [1–4] which was performed using numerical methods for Koopman operator approximation that are closely related to those used in the diagnostic study in this paper. Other cited prognostic studies include prediction of Arctic sea ice using Dynamic Mode Decomposition (DMD) [5] and tropical pacific SST using kernel approximations of transfer operators [6].

3. *Concerning the MPT, the paper only mentions the role of CO2 and regoliths in its occurrence and proposes its own minimal model F as the basis of the analysis. Riechers et al. (Clim. Past, 2022) have reviewed in extenso the issue of intrinsic climate oscillations and their role in the MPT, in the context of nonautonomous dynamics. Their paper’s Table A1 extends over 3 pages and tenths of models, both with and without intrinsic oscillations. Can this paper’s methods help resolve, for instance, the issue of whether the MPT has been caused by sub- or supercritical Hopf bifurcation?*

Response: Model F is a highly idealized model that is primarily meant to replicate frequency-switching behavior reminiscent of glaciation cycles rather than directly model glaciation cycles. We have added text surveying papers on modeling and analysis of glaciation cycles and the MPT on p. 19. This text includes references to Riechers et al. and the SSA work of Vautard & Ghil 1989. We would rather not speculate about possible connections between our approach and bifurcation analysis.

With regards to connections with SSA, since we have added extensive material in the manuscript on SSA, we have also added a comparison between the Koopman/transfer operator eigenfunctions and results from SSA in Supplementary Note 3 and Supplementary Figures 3 and 4 (the latter, reproduced as Fig. R1 below). The SSA modes do not appear to separate the trend, the 40 kyr cycles, and the 100 kyr cycles into individual modes with the same level of fidelity as the Koopman/transfer operator eigenfunctions, and thus would be a less effective basis for reconstructing the salient features of the MPT.

4. *SSA is based on K-L theory, which requires at least weak stationarity of a random process to apply. Eckmann & Ruelle (RMP, 1985) have told us a long time ago that spectral theory can apply to deterministically chaotic dynamics, too. But SSA has been applied quite successfully to nonstationary time series, whether deterministically or stochastically generated; see the Ghil & Vautard (1991) results above as one of very many such applications. In fact, SSA typically produces a nonlinear trend as the leading component of the decomposition in such cases; here, nonlinear means that the corresponding $x_1 = x_1(t)$ graph is not a straight line. It might be quite interesting to understand these “facts of life” better by using a novel and ingenious analogy or actual mathematical relationship between “Koopmania” and SSA. Santos Gutiérrez et al. (Chaos, 2021) have, in this spirit, helped understand the remarkable success — both diagnostic and prognostic — of empirical model reduction, originally a purely data-driven method, in the light of analytical, operator-theoretical methods.*

Response: We have added a discussion on the connection between eigenfunctions of a pure Koopman/transfer operator and multichannel SSA on p. 12 and discussed Ghil & Vautard (1991) (see response to point 2 above). We elaborate on this connection in more technical terms below. However, we emphasize that the diffusive processes we add in the current work are crucial to the success of autonomous operator-theoretic methods obtaining trends and cycle-switching. The diffusion theoretically explains the form of the eigenfunctions in Models M, A, and F, and is computationally necessary to correctly order the dominant eigenvalues by magnitude. These considerations are absent from SSA and Koopmania.

Connections between SSA and Koopman/transfer operator methods: We have studied connections between kernel methods employing delay-coordinate maps (of which SSA is a special case utilizing the covariance kernel) and Koopman operator eigendecomposition in previous work [7, 8]. To give an outline of the conclusions stemming from that work, suppose that one has $N + M$ time-ordered observations $x_1, \dots, x_{N+M} \in \mathbb{R}^d$ coming from a measure-preserving, ergodic system; that is, there is a state space Ω , a measure-preserving transformation $T : \Omega \rightarrow \Omega$ with invariant measure μ and an observable $F : \Omega \rightarrow \mathbb{R}^d$ such that $x_n = F(\omega_n)$ with $\omega_n = T^n(\omega_0)$ for some initial condition $\omega_0 \in \Omega$. In multi-channel SSA with M delays, one computes the SVD of an $(Md) \times N$ Hankel matrix of the form

$$\mathbf{X}_M = \begin{pmatrix} x_1 & \cdots & x_N \\ x_2 & \cdots & x_{N+1} \\ \vdots & & \vdots \\ x_M & \cdots & x_{N+M} \end{pmatrix}. \quad (1)$$

The right singular vectors of \mathbf{X}_M can be equivalently computed as eigenvectors of the $N \times N$ covariance matrix $\mathbf{C}_M = \mathbf{X}_M^\top \mathbf{X}_M / M$. One can then view \mathbf{C}_M as an approximation of an integral operator $C_M : L^2(\Omega, \mu) \rightarrow L^2(\Omega, \mu)$,

$$C_M f = \int_{\Omega} c_M(\cdot, \omega) f(\omega) d\mu(\omega),$$

where $c_M : \Omega \times \Omega \rightarrow \mathbb{R}$ is given by

$$c_M(\omega, \omega') = \frac{1}{M} \sum_{m=1}^M F(T^m(\omega)) \cdot F(T^m(\omega')).$$

[We realize, of course, that the Reviewer is an expert on this topic; we mainly went through this construction to fix notation.]

Next, let $U : L^2(\Omega, \mu) \rightarrow L^2(\Omega, \mu)$ be the Koopman operator associated with T , i.e., $Uf = f \circ T$. Consider the commutator $A_M = [C_M, U]$. It can be shown [7] that as $M \rightarrow \infty$, (i) C_M has a well-defined limit, C_∞ , in the operator norm topology, and C_∞ is a compact operator; (ii) the commutator A_M converges to 0 in operator norm. Put together, these two facts imply that the eigenspaces of C_∞ corresponding to nonzero eigenvalues (i.e., the temporal patterns that one asymptotically recovers from multi-channel SSA for infinitely many samples N and infinitely many delays M) are finite orthogonal sums of eigenspaces of U (which are one-dimensional by ergodicity). In other words, so long as the range of C_∞ contains non-trivial (i.e., non-constant) Koopman eigenfunctions, ‘‘Koopmania’’ and SSA can yield asymptotically equivalent eigenfunctions. We believe that this provides a rather direct correspondence between the two methods.

With that said, we would like to stress that the SSA and transfer/Koopman operators are usually applied to finite data sets and in this situation, several differences appear.

1. The fact that C_∞ and U have common eigenfunctions has no bearing on how *many* Koopman eigenspaces are contained in the range of C_∞ . In particular, C_∞ may be a finite-rank operator and thus able to represent only a limited number of Koopman eigenspaces.
2. For any fixed $M < \infty$ the rank of C_M is at most Md . Aside from special cases (e.g., the observation map F has a finite expansion in terms of Koopman eigenfunctions), this places a significant limitation on the number of Koopman eigenfunctions that can be exactly recovered, even in the infinite-data limit ($N \rightarrow \infty$). A related issue occurs in the case of the Hankel-EDMD technique which can be understood as Koopman operator approximation in a dictionary of observables given by the eigenvectors of C_M

(see our response to comment 4 of Reviewer 2). In contrast, our approach for Koopman operator approximation employs dictionaries obtained from eigendecomposition of kernel integral operators that have *infinite* rank even when the data space dimension d and number of delays M are finite. In broad terms, we use dictionaries obtained from operators of the form

$$K_M f = \int_{\Omega} k_M(\cdot, \omega) f(\omega) d\mu(\omega)$$

where k_M is a heat-like Markovian kernel that can be thought of as a nonlinear generalization of c_M . By appropriately choosing k_M , the operator K_M has infinite rank and its data-driven approximation from N samples has rank N , allowing our approach to recover an arbitrarily large number of Koopman eigenspaces as the amount of available data increases.

3. The transfer/Koopman operator spectrum provides direct timescale information for each eigenfunction from the corresponding eigenvalue. Obtaining characteristic timescales from the eigenvectors of C_M requires postprocessing (e.g., Fourier analysis).
4. As discussed in detail in the manuscript using the idealized models M, A, and F as examples, when analyzing a single time series generated by an autonomous system it is beneficial to perturb the transfer/Koopman operators by diffusion. Effectively, the diffusion process allows the perturbed operators to capture geometrical information of the non-autonomous state space trajectory through eigenfunctions. The heat-like operators K_M provide a natural means of implementing this diffusive perturbation of the dynamics numerically. However, it is not obvious how to perform a similar construction using covariance operators (e.g., the covariance kernel $c_M(\omega, \omega')$ is not sign-definite).

Discussion and recommendation. The list of Major Comments above is clearly an ambitious one, justified by the quality of the paper at hand, in its current version. I think that the paper should certainly be published in a Nature Publishing Group journal. Which one should be determined by the improvements that the authors will choose to address in this paper, as opposed to some future ones. With this recommendation in mind, I've added a small selection of Minor Comments as an Appendix to this review.

Response: We believe that the new material added in response to the Reviewer's comments has strengthened the paper, and helped better position it in the context of the literature and highlight its novel aspects. We hope that our revision has adequately addressed both major and minor comments of the Reviewer.

p. 2, 1st full para. Benthic $\delta^{18}O$ radioisotope concentration is a proxy for global temperature and ice volume (e.g., Lisiecki & Raymo, Paleo, 2005, p. 2, para. [5]), not of atmospheric CO_2 concentration!

Response: Thank you, we have amended the text to say that $\delta^{18}O$ is a proxy of ambient seawater temperature.

p. 4, 1st full para. The special circular arrow notation for a map T from Ω to itself is sexy, & I've happily used it myself at the blackboard. But it is replaced later in the paper with the more common $T : \Omega \rightarrow \Omega$ notation. Maybe use only the latter one?

Response: We have replaced the circular arrows with straight arrows.

p. 8, 1st l. after Eq. (4): Explain a bit "row-stochasticize"; maybe add an appendix or two to explain the more technical aspects of the paper to a wider climate-interested audience.

Response: We have replaced row-stochasticize with row-normalize; the operation had already appeared in the following equation (5). The technical aspects are explained to a level we felt appropriate for the audience in the Methods section, with references to Froyland/Giannakis/Lintner/Pike/Slawinska (2021) as needed. We felt this is a better approach than duplicating material in an appendix, which would have to appear in the Supplementary Information due to page restrictions.

p. 14, Fig. 8. The panel labeling in the figure caption is not consistent with the text and the panel legends.

Response: Thanks for catching this. Caption has been corrected.

p. 15, Fig. 9. This figure & Fig. 8 are the most striking, & only truly novel, results of the application of the methods to the two climate problems. It might be worthwhile including in the possible explanations a weakening of the Pacific South American (PSA) anomaly pattern (Mo & Ghil, JAS, 1987; Li & Le Treut, GRL, 1999).

Response: Thank you for this suggestion. As you have pointed out, the PSA is typically associated with the SACZ, so trend-related changes of the SACZ may be related to trend-related changes of the PSA. We have added relevant citations and discussion on p. 15. Since precipitation changes can also be due to other factors (e.g., changes of tropical circulation patterns such as low-level jet and Bolivian jet, and general influence of Atlantic Ocean), other possible explanations and references have been added as well [9, 10].

p. 17, Fig. 10. Panels (b)–(e) look amazingly like SSA results; see, for instance, Fig. 2 in Kondrashov et al. (GRL, 2005) for an oscillatory component or Fig. 4 of Moron et al. (Clim. Dyn., 1998) for industrial-era trends.

N.B. In SSA, the eigenvector associated with a trend has no sign change but is not identically equal to 1 (different version of the Perron-Frobenius theorem or some version of Sturm-Liouville theory); it is a data-adaptive set of positive weights that is likely to yield a better running mean than the operator-theoretic one. A closer look might be worth it.

Response: While panels (b)–(e) in Figure 10 may look superficially similar to Figure 2 in Kondrashov et al., the former is computed from benthic $\delta^{18}\text{O}$ data sampled every 1000 years over millions of years, while the latter is computed from Nile River water level record data sampled annually over hundreds of years, so any correspondence is coincidental. Similarly, the industrial-era trends from Fig. 4 of Moron et al. are not related to Figure 10. We have included a citation to Moron et al. in our discussion of industrial-era trends on pp. 13–14.

2 Response to Reviewer #2

The authors provide an application of Koopman and transfer operator frameworks for autonomous systems to analysis of non-autonomous systems. They then apply the framework to two real-world examples from present and past climate dynamics: Variability of sea surface temperature (SST) over the industrial era and the mid-Pleistocene transition (MPT) of Quaternary glaciation cycles. Separating forcing and internal variability aspects is, of course, a valuable endeavor. But it is not clear to me how much is new here on the methodological side, because the paper is quite sparse on referencing previous work by others on analysis of nonautonomous systems using Koopman operators. Here are some examples and additional comments.

Response: We emphasize that the main contributions of the submission do not include the development of new methods to approximate Koopman and transfer operators; indeed the implementations used in the submission have already appeared in the literature [11]. The main novelty lies in the interpretation of the eigenfunctions of stochastically perturbed transfer and Koopman operators, built using standard autonomous approaches from a *single* nonstationary time series. Leading eigenfunctions can extract trends and frequency-switching from the time series, and until now, this has not been investigated and explained in detail. Moreover, we provide a geometrical interpretation for why time-delay embedding methodologies can extract distinct frequency components from time series with switching frequency (Lemma 1).

1. Page 1. There are a number of missing references on analysis non-autonomous systems using operator theoretic techniques: by typing e.g. “koopman and nonautonomous” into Google Scholar, we find Mezic and Surana (2016) is perhaps the first one, but many followed. What should be reviewed in addition is work on system identification for controlled systems (which are nonautonomous by definition), see eg Mauroy et al book “Koopman operator in systems and control”.

Response: The main purpose of this work is to explain why naive application of autonomous transfer and Koopman operator methods to nonstationary time series data can produce useful trend or switching

information through their dominant eigenvectors, when there is no good *a priori* reason for this. A second crucial aspect of this work is the use of a *single* time series as input data. Considering the main two aspects of this work, we found Mezić/Surana (2016) largely orthogonal to our submission; it does not discuss or explain trends and it uses multiple trajectories. Similarly, the setting of system identification for controlled systems is one where one typically has access to multiple actuated trajectories. That being said, we agree that discussing methods utilizing multiple trajectories is appropriate for placing our work in context. In this revision, we have included a discussion and citations of techniques utilizing multiple trajectories, including Mezić and Surana (2016) and Mauroy et al., on p. 2. We have also added a reference to the paper [5] from the Mezić group on sea ice prediction (see p. 12).

2. Page 2, “We will show in this present work that an estimate of the (non-stationary) Indo-Pacific warming trend, as well as the modulation of the seasonal cycle by the trend, can be obtained through eigenvectors of transfer or Koopman operators.”. Isn’t it clear how to do this since Mezić-Surana? Namely, a linear trend can be captured by adding a variable L describing that trend with dynamics $\dot{L} = 0$. Any cyclic forcings can also be captured that way, by using the notion of “suspension”. The authors use time-delays here, but that is not novel either - see the comment below on Hankel-DMD.

Response: The trend is unknown *a priori*, so it is not clear how adding a variable L would pick up this unknown trend. The suggestion sounds like time-expansion, if indeed $\dot{L} = 1$ is meant rather than $\dot{L} = 0$. If $\dot{L} = 1$ is meant, why choose the number 1 rather than 1000? The result will be very sensitive to the choice of 1 vs 1000. More broadly, introducing such a scheme would destroy crucial geometric relationships in the data that would otherwise be present, especially when there is only a single time series.

3. Pg 3, Figure 1, this kind of dynamics, and the application of autonomous Koopman operator methods to the nonautonomous fluid flow around a cylinder, can be found in Glaz et al (2017) paper “Quasi-periodic intermittency in oscillating cylinder flow”

Response: Glaz et al. (2017) discusses a periodically forced fluid flow with a fixed forcing frequency. This is different to the dynamics illustrated in the idealized models in Figure 1, which undergo frequency switching at random or unpredictable times. We could not find work on frequency switching in Glaz et al. (2017).

4. Pg 5 “A classical approach to such a time series might be to embed it in a higher-dimensional space using Takens method of delays”. The paper does not point out a number of relevant references: there is a history of using time delays in Koopman operator context starting with Mezić and Banaszuk (2004), and extensions to EDMD were done in Arbabi and Mezić (2017) - the later under the name of Hankel-DMD, which is in massive use these days. Is there any difference with the methods developed here to approximate operators?

Response: The main theoretical ideas in the submission are universal and independent of any specific numerical approximation technique. One may use any quality method of approximating transfer or Koopman operators with added diffusion. We use the Takens method of delays only to improve state estimation; it is not an important aspect of the general theory in our submission. If we were provided with a nonstationary trajectory of full states, then we would not use delays at all when constructing our operators.

Our operator-approximation techniques make use of regularization, either by composition of the transfer operator with heat-like kernel integral operators, or by addition of diffusion to the generator. This step is important for ensuring well-conditioned approximations even in the autonomous case (e.g., when the Koopman operator has continuous spectrum or a dense set of eigenvalues), and it is especially important in the context of non-autonomous systems studied in this work. As discussed in the paper, the eigenfunctions approximated from a *single* time series depend on a balance between deterministic dynamical operators and diffusion operators associated with the geometry of the embedding of the dynamical trajectory in delay-embedding space.

With regards to Arbabi and Mezić (2017) [12], we have carefully reviewed the paper and attempted to analyze the $\delta^{18}\text{O}$ data using their algorithms and the code in the repository <https://github.com/arbabiha/DMD-for-ergodic-systems>. However, to the best of our understanding, the methods of [12] do not output approximations of Koopman eigenfunctions; instead, they compute Koopman *modes* (projections of Koopman eigenfunctions onto the observed data). Specifically, the output of these algorithms applied to the Hankel matrix (1), which is dimensioned $Md \times N$, is to produce eigenvectors (Koopman modes) of dimension Md

and not N . This means that without additional processing, the algorithms of [12] cannot be used to produce eigenfunction time series analogous to Fig. 10 in the manuscript and Fig. 3 in the Supplementary Information because the corresponding eigenvectors would need to have dimension N .

We are unsure what [12] had in mind for computation of Koopman eigenvectors, but we believe that a closely related approach that does produce approximations of Koopman eigenfunctions is to use the basis of right singular vectors of the Hankel matrix as a dictionary for EDMD. This appears to be closely related to the ‘‘SVD-Enhanced DMD’’ algorithm described in [12] with the difference that using right singular vectors of the Hankel matrix produces N -dimensional Koopman eigenvectors whereas SVD-Enhanced DMD produces (dM) -dimensional vectors (Koopman modes).

In Fig. R2 we show results obtained with this approach applied to the $\delta^{18}\text{O}$ data analyzed in the paper. We order the EDMD eigenfunctions, v_j^{EDMD} , in order of decreasing modulus of the corresponding eigenvalue, λ_j^{EDMD} . For the eigenvalues with nonzero imaginary part we compute corresponding periodicities $T_j = 2\pi(\Delta t)/(\text{Im} \log \lambda_j)$ where $\Delta t = 1$ kyr is the sampling interval of the data.

The leading EDMD eigenvector, v_1^{EDMD} (Fig. R2(b)), exhibits trend-like behavior, but it is noisier than the corresponding eigenvector in Fig. 10(b) of the paper and Fig. 3(b) of the SI, and appears to mix oscillatory behavior into the post-MPT period (last 1 million years; see the wavelet spectrum in Fig. R2(j)). The other EDMD eigenvectors shown in Fig. R2(c–e) exhibit oscillatory behavior on timescales that are interpretable in terms of ~ 40 kyr, ~ 20 kyr, and ~ 100 kyr orbital forcings (see p. 16 in the manuscript). However, the frequency switch associated with the MPT is not discernible in these eigenfunctions.

Besides these results, the difference between Hankel–EDMD and our operator approximations in terms of theory vs. practice is discussed in point 2 of the response to query 4 of Reviewer 1. In particular, we perform operator approximation in an eigenbasis of kernel operators associated with nonlinear, Markovian kernels—these operators have infinite rank (unlike the covariance operator C_M) and their range is dense in L^2 which allows the methods to converge without making the restrictive assumption that the observed data lie in finite-dimensional Koopman-invariant subspaces (see, e.g., [12, Proposition 5] and related results).

In summary, I do think this is an interesting topic, but it is unclear what aspects of methodology are actually new. I am open to reconsidering upon author’s explanation.

Response: At the heart of the submission is the theoretical explanation for why naive computation of dominant eigenvectors of a transfer or Koopman operator from nonstationary data may numerically show meaningful trends. Strictly speaking, such a computation does not make any sense because eigenvectors implicitly assume autonomicity. We show how with the right combination of diffusion, the state-space geometry creates a Markov diffusion process that approximates the action of the transfer/Koopman operator, and has dominant eigenfunctions that can extract trends in the original nonstationary data. Other novel aspects include a geometric lemma and the introduction of three classes of fundamental idealized models to illustrate and explain the core ideas. The $\delta^{18}\text{O}$ data provides a well-studied validation of our novel ideas and approach, while providing higher-fidelity eigenfunctions than EDMD or SSA. The SST data leads to truly novel results for precipitation trends in South America.

3 Response to Reviewer #3

In their new manuscript, Froyland et al. present theoretical foundations and climate applications of a new operator-theoretic framework for decomposing time series into trends and persistent cycles, which radically differs from most established statistical time series decomposition techniques. There is a great level of novelty in this interesting work, and I may foresee vast potentials for further applications across various fields of science.

The paper is essentially well written, although I tend to struggle a little bit with the distinction between results and methods sections as requested by the journal’s native manuscript format (for example, one may argue if all of the section on time-delay embedding fits into the results section). Also the order of figures and their referencing across the text does not always appear fully logical to me. However, I still think the authors did a good job given the corresponding requirements.

The few concerns and suggestions for improvements that I do have (listed below) amount to a recommendation for acceptance of this interesting work after minor revision.

We thank the Reviewer for their positive assessment of our work.

1. The operator based techniques developed and employed in this work are related to “eigenfunctions of Koopman and transfer operators” (abstract, l.12), but it is not quite clearly worked out that is it actually the transfer operator that is exploited in this work (if I understand this point correctly). I suggest revising the text accordingly, or alternatively clarify briefly the link between the eigenfunctions of both operators.

Response: Transfer operator computations are illustrated in the Model M, A, and F sections. Because of space limitations, the transfer operator computations for the SST trend and the $\delta^{18}\text{O}$ frequency switching appear in the Supplementary Information. An extended figure for the transfer operator $\delta^{18}\text{O}$ eigenfunctions has been included as Supplementary Fig. 3.

In the “Eigenfunctions for Model M”, “Eigenfunctions for Model A”, and “Eigenfunctions for Model F” sections (pp. 8–11) we had previously (i) discussed the specific matrix P , which is an approximation of the transfer operator (see the section “Transfer and Koopman operator computation”) and (ii) talked about forward time steps due to positive s . Each of these aspects refers to the transfer operator, but in fact, the analysis equally applies to the Koopman operator, which would correspond to negative s . We have added a sentence of clarification in the “Transfer and Koopman operator computation” section (p. 8) and at the end of the “Eigenfunctions for Model M” section (p. 9) to further highlight that the transfer operator is used in the computations for Models M, A, and F, and that the Koopman operator could be equally used instead.

2. The concept of a trend is a somewhat ambiguous thing in statistics. My understanding is that it describes a specific component embedded in a time series that behaves completely monotonically over the considered observation period. While this may merely sound a subtlety, it may be relevant in interpretations of climate applications, where a wide audience may associate “trend” with anthropogenic effects beyond natural variability. While the presented technique seems powerful to identify such monotonic components in climate time series, any attribution of the so-called trend component to actual trends (long-term changes beyond the period of observations) versus, for example, multi-decadal variability may be problematic. My suggestion would be clarifying the concept of a “trend” in the context of the present work being a statistical one, not necessarily implying a trend in the sense of the common use of this term in climate sciences.

Response: We have added text in the introduction (p. 2) to clarify that we are talking about a statistical trend (if we understand the Reviewer’s comments correctly, by definition any trend computed on the basis of observations alone is deemed to be a statistical trend).

3. The terminology used with respect to the paleoclimate example is incorrect. $\delta^{18}\text{O}$ is not a radioisotope concentration, but the ratio between the respective concentrations of two oxygen isotopes in the studied material (O-18 over O-16). The benthic $\delta^{18}\text{O}$ stacked record studied here should also not be interpreted as a proxy of atmospheric CO_2 concentrations, but rather as representative of ambient water temperatures (while also being affected by water salinity and global ice volume). There exists a potential indirect dependency on atmospheric CO_2 concentrations, which is however non-trivial.

Thank you for noticing our inaccurate usage of terminology. We changed the text to say “paleo-proxy for ambient seawater temperature”. We also changed “radioisotope concentration” to “oxygen isotope ratio”.

*4. Regarding the paleoclimate application, it might be interesting to have a brief discussion on how the newly proposed technique behaves in comparison with other established statistical techniques like continuous wavelet transform, singular spectrum/system analysis, or empirical mode decomposition. I do not expect any additional analyses on this aspect, but would welcome some general perspectives on this from the authors, potentially along with references to published works (e.g. Nyman & Ditlevsen, *Climate Dynamics*, 2019; Mukhin et al., *Scientific Reports*, 2019). Some reference may be useful to the statement on the last sentence before the discussion (top of page 18: “...known to occur”).*

Response: The results in Fig. 10 already include continuous wavelet transforms. We have added some remarks and the suggested references on p. 19 on distinctions between our operator eigendecomposition

approach, SSA and, Fourier/wavelet methods. Unlike Fourier/wavelet methods, which involve additional choices of spectral processing/filtering, operator eigendecomposition yields an independent collection of eigenfunctions. These eigenfunctions can be close in the frequency domain, which would make identifying them by spectral filtering/windowing challenging, but they are automatically recovered as orthogonal functions using our approach. For example, the trend combination eigenfunctions used in our reconstruction of South American precipitation have almost identical frequency to the seasonal-cycle modes, but they are well-separated in terms of the real part of the corresponding eigenvalue in the complex plane (see the spectrum in Supplementary Figure 1), and they are orthogonal as complex-valued time series (equivalently, orthogonal as L^2 functions on state space with respect to the empirical sampling measure). On p. 19, we have included a brief discussion on statistical and state-space methodologies, including the references suggested.

As a brief additional analysis, we included Supplementary Note 4 in the SI and a figure (Supplementary Fig. 5) on reconstructions of the $\delta^{18}\text{O}$ time series using the eigenfunctions identified in Fig. 10. The results demonstrate the efficiency of the eigenfunctions computed with our approach to reconstruct the salient features of $\delta^{18}\text{O}$ pertaining to Quaternary glaciation cycles and the MPT. Note, in particular, that the reconstructed $\delta^{18}\text{O}$ time series automatically have “clean” wavelet spectra without requiring processing in time–frequency domain.

5. On the interpretation of the eigenfunction families for the Indo-Pacific SST data: The authors associate their findings to “meridional shifts of the South American monsoon”. I am wondering if it is indeed the monsoon or rather the inter-tropical convergence zone that is shifting here. It may be worth looking a bit more into the seasonality of the underlying patterns. I would also be wondering if the proposed explanation of Fig. 8 in terms of Hadley cell and SACZ intensifications (bottom of page 15) can be supported by any reference to existing climatological studies.

Response: Thank you for this suggestion. We have modified the text to reflect the fact that ITCZ meridional shifts constitute a primary mechanism (and interactions with land over subtropical regions can modify SAM). The difference between ITCZ and monsoon can be subtle (see, e.g., definitions in the second paragraph of [13]). Our regional analysis (in both the tropical belt and subtropics) is based on time-evolving snapshots of reconstructed precipitation, thus our claims are based on analysis of seasonality changes of precipitation (and we show two representative snapshots in Fig. 8). We have also added a citation related to regional intensification of the Hadley Cell and SACZ intensification [14].

Minor comments

- *It is not completely obvious to me what “slowly decaying variables” should be (already in the abstract, l.9) – maybe two or three additional words could clarify this.*

Response: We could find two instances of “slowly decaying” (one in the abstract and one in the introduction). We agree this could be confusing and so we have changed or added the text “slowly decorrelating”.

- *Abstract: I suggest briefly mentioning the type of data (reanalysis, paleoclimate proxies) used in the two real-world case studies.*

Response: That’s a good idea, we have made the change in the abstract.

- *Page 5, l.2: “The same dynamical model T as in (1)...”*

Response: Thank you, we have corrected the typo.

- *Figure 6, caption: is it really two complex eigenvectors or two pairs of eigenvectors with complex-conjugate eigenvalues? (similar shortages may be found in other places of the text)*

Response: Thank you, the description was previously too loose. We have now been more specific in several places by discussing complex-conjugate pairs instead of individual eigenvalues.

- *Page 14: I think “austral” is commonly not capitalized in climate studies when referring to “southern hemispheric”.*

Response: We found three instances in the manuscript and uncapitalized them.

- Page 15, bottom: when referring to a point in deep time, one might commonly use, e.g., “1 Myr BP (before present)”, as opposed to references to time spans (“spanning 2.6 million years (Myr)” – here “ago” should be removed).

We changed the text to “extending from ~ 2.6 million years ago (Ma) to the present”. We also changed Mya to the more commonly used Ma throughout the manuscript and in Fig. 10.

- Page 20, Generator approximation: Should $\text{Im}\omega_i$ rather be $\text{Im}\gamma_i$ (which might be what the authors refer to as ω_i)?

Response: That’s correct; thanks for catching the typo. Change made.

- Page 23, ref. 25: check author

Response: Fixed, thank you.

- Page 23, ref. 26: typo in “feedback”

Response: Fixed, thank you.

- Suppl. Fig. 1: I think here is an inconsistent notation being used as compared to the main text (eigenvalues denoted by λ_i instead of γ_i)

Response: Yes, the notation in Supplementary Fig. 1 was inconsistent. We changed λ_i to γ_i .

References

- [1] R. Alexander, Z. Zhao, E. Szekely, and D. Giannakis. Kernel analog forecasting of tropical intraseasonal oscillations. *J. Atmos. Sci.*, 74:1321–1342, 2017. doi:[10.1175/JAS-D-16-0147.1](https://doi.org/10.1175/JAS-D-16-0147.1).
- [2] N. Chen, A. J. Majda, and D. Giannakis. Predicting the cloud patterns of the Madden-Julian Oscillation through a low-order nonlinear stochastic model. *Geophys. Res. Lett.*, 41(15):5612–5619, 2014. doi:[10.1002/2014gl060876](https://doi.org/10.1002/2014gl060876).
- [3] D. Comeau, D. Giannakis, Z. Zhao, and A. J. Majda. Predicting regional and pan-Actic sea ice anomalies with kernel analog forecasting. *Climate Dyn.*, 52(9–10):5507–5525, 2019. doi:[10.1007/s00382-018-4459-x](https://doi.org/10.1007/s00382-018-4459-x).
- [4] X. Wang, J. Slawinska, and D. Giannakis. Extended-range statistical ENSO prediction through operator-theoretic techniques for nonlinear dynamics. *Sci. Rep.*, 10:2636, 2020. doi:[10.1038/s41598-020-59128-7](https://doi.org/10.1038/s41598-020-59128-7).
- [5] J. Hogg, M. Fonoberova, and I. Mezić. Exponentially decaying modes and long-term prediction of sea ice concentration using Koopman mode decomposition. *Sci. Rep.*, 10:16313, 2020. doi:[10.1038/s41598-020-73211-z](https://doi.org/10.1038/s41598-020-73211-z).
- [6] A. Navarra, J. Tribbia, and S. Klus. Estimation of Koopman Transfer operators for the equatorial Pacific SST. *J. Climate*, 78(4):1227–1244, 2021. doi:[10.1175/JAS-D-20-0136.1](https://doi.org/10.1175/JAS-D-20-0136.1).
- [7] S. Das and D. Giannakis. Delay-coordinate maps and the spectra of Koopman operators. *J. Stat. Phys.*, 175(6):1107–1145, 2019. doi:[10.1007/s10955-019-02272-w](https://doi.org/10.1007/s10955-019-02272-w).
- [8] D. Giannakis. Delay-coordinate maps, coherence, and approximate spectra of evolution operators. *Res. Math. Sci.*, 8:8, 2021. doi:[10.1007/s40687-020-00239-y](https://doi.org/10.1007/s40687-020-00239-y).
- [9] I. F. A. C. Cavalcanti and M. H. Shimizu. Climate fields over South America and variability of SACZ and PSA in HadGEM2-ES. *Am. J. Clim. Change*, 1(3):132–144, 2012. doi:[10.4236/ajcc.2012.13011](https://doi.org/10.4236/ajcc.2012.13011).
- [10] F. A. Cavalcanti, N. J. C. Barreto, M. S. Alvarez, M. Osman, and C. A. S. Coelho. Teleconnection patterns in the Southern Hemisphere represented by ECMWF and NCEP S2S project models and influences on South America precipitation. *Meteorol. Appl.*, 28(4):e2011, 2021. doi:[10.1002/met.2011](https://doi.org/10.1002/met.2011).

Figure R1: Leading right singular vectors (temporal modes) from SSA applied to the $\delta^{18}\text{O}$ data. The results are plotted in a similar manner to Fig. 10 in the manuscript. To create the 2D trajectories in panels (g)–(i) we used pairs of singular vectors with approximately equal corresponding singular values.

- [11] G. Froyland, D. Giannakis, B. Lintner, M. Pike, and J. Slawinska. Spectral analysis of climate dynamics with operator-theoretic approaches. *Nat. Commun.*, 12:6570, 2021. doi:[10.1038/s41467-021-26357-x](https://doi.org/10.1038/s41467-021-26357-x).
- [12] H. Arbabi and I. Mezić. Ergodic theory, dynamic mode decomposition and computation of spectral properties of the Koopman operator. *SIAM J. Appl. Dyn. Sys.*, 16(4):2096–2126, 2017. doi:[10.1137/17M1125236](https://doi.org/10.1137/17M1125236).
- [13] R. P. Acosta, J.-P. Ladant, J. Zhu, and C. J. Poulsen. Evolution of the Atlantic Intertropical Convergence Zone, and the South American and African monsoons over the past 95-Myr and their impact on the tropical rainforests. *Paleoceanogr. Paleoclimatol.*, 37(7):e2021PA004383, 2022. doi:[10.1029/2021PA004383](https://doi.org/10.1029/2021PA004383).
- [14] D. Kim, H. Kim, S. H. Kang, M. F. Stuecker, and T. Merlis. Weak Hadley cell intensity changes due to compensating effects of tropical and extratropical radiative forcing. *NPJ Clim. Atmos. Sci.*, 5:61, 2022. doi:[10.1038/s41612-022-00287-x](https://doi.org/10.1038/s41612-022-00287-x).

Figure R2: As in Fig. R1, but for eigenvectors computed using Hankel EDMD.

REVIEWERS' COMMENTS

Reviewer #1 (Remarks to the Author):

Review of the Revised version and Rebuttal of

Froyland et al. "Revealing trends and persistent cycles [...]"

General and Recommendation. The authors have made a substantial and thorough effort at addressing the issues raised in my first review. The current version is greatly improved and should be published in Nature Communications. The points below are relatively minor and are only meant to improve the final, published paper even more.

Semi-major comment

The authors' effort in Supplementary Note 3 of carrying out a comparison between SSA and their Koopman/transfer operator–based technique is quite praiseworthy. As is often the case in such comparisons, the authors' method wins. In the case at hand, the authors only used the "vanilla" version of SSA, as introduced in the Vautard & Ghil (Physica D, 1989) and Ghil & Vautard (Nature, 1991) papers. In such an implementation, the separation of trend and distinct oscillatory components is not optimized. More recently, Groth & Ghil (PRE, 2011; J. Clim., 2015)

have shown that this problem can be fixed by appropriate rotations of the eigenvectors and illustrated the cleaner results thus obtained on very large climatic (Groth et al., J. Clim., 2017) and macroeconomic (Groth & Ghil, Chaos, 2017) data sets.

Clearly, I do not insist that the authors turn their attention to these finer points of SSA methodology. But it would be nice to mention that such improvements do exist and provide the appropriate references.

Minor comments

1. It would be good if the authors devised a short name and an acronym for their method, to better place it in the context of other methods.
2. Likewise, a shorter title for the paper would attract greater attention to the work therein.
3. The reference to Mo & Ghil (JAS, 1987) in the current version's ref. [72] is incomplete since it does not include the name of the 2nd author.

4. I'm not sure that I understand the sentence on p. 12, blue para., "This provides an approximate interpretation of the cyclicity of the dominant modes extracted by SSA using many delays." I thought that the embedding in SSA and the authors' method is the same Mañe-Takens embedding.

5. Slight correction to a sentence on p. 15, 1st para., "These patterns explain a relatively small amount of variance of the raw Indo-Pacific SST data (about two orders of magnitude smaller than the seasonal cycle pair {v1,v2}) and, as we have verified in separate calculations, are not captured in the SSA/EOF spectrum (possibly since SSA/EOFs optimize for explained variance [...])" It is only the vanilla version mentioned above that orders EOFs by variance. Neither Monte Carlo SSA (Allen & Smith, *J. Clim.*, 1996) nor the previously mentioned references do.

6. There are several precursors to SSA and multichannel SSA (M-SSA) and we have done our best to mention all those we knew about in the Ghil et al. (*Rev. Geophys.*, 2002) review paper. I wouldn't pick Weare & Nastrom's (MWR, 1982) paper as the most interesting or visionary one of them: their EOFs were only extended to three successive maps and there was no indication of embedding, determining cyclicity, etc.

Conclusion. It's a pleasure to congratulate the authors on a fine paper and I look forward to its publication in *Nature Communications*.

Additional references

1. Allen, M.R. and L.A. Smith, 1996: Monte Carlo SSA: Detecting irregular oscillations in the presence of colored noise. *J. Climate*, 9(12), 3373–3404.

2. Groth, A., and M. Ghil, 2011: Multivariate singular spectrum analysis and the road to phase synchronization, *Phys. Rev. E*, 84, 036206 (10 pp.), doi:10.1103/PhysRevE.84.036206.

3. Groth, A., and M. Ghil, 2015: Monte Carlo singular spectrum analysis (SSA) revisited: Detecting oscillator clusters in multivariate data sets, *J. Climate*, 28, 7873–7893.

4. Groth, A., Y. Feliks, D. Kondrashov, and M. Ghil, 2017: Interannual variability in the North Atlantic ocean's temperature field and its association with the wind-stress forcing, *J. Climate*, 30, 2655–2678, doi: 10.1175/JCLI-D-16-0370.1

5. Groth, A., and M. Ghil, 2017: Synchronization of world economic activity, *Chaos*, 27, 127002 (18 pp.), doi: 10.1063/1.5001820.

Reviewer #3 (Remarks to the Author):

I have very much enjoyed re-reading this inspiring manuscript after revision. In my opinion, the authors have done a good job in addressing the comments of all reviewers, except for some minor oversights that I will list below along with some additional observations of mine suggesting a few last corrections and clarifications.

Minor comments:

- P.2, ll.24-25: It might help clarifying why the authors suggest a harmonic model to capture both a quite periodic (seasonal) and a rather aperiodic (ENSO) cyclic variation. I suppose that this is just meant as a zero-order “conceptual” model, so why not state this explicitly?
- P.3, caption of Fig. 1, ll.10-13: The wording and referencing in those lines is a bit confusing with two sentences of almost identical wording but opposite statements. This may be partially because there is no explicit referencing to the lower panel in the caption. Please revise the caption accordingly.
- As already emphasized in my original review, $\delta^{18}\text{O}$ is a ratio of the concentrations of two different oxygen isotopes, not a concentration itself. This is still stated erroneously at a couple of places within the manuscript (and needs corresponding correction): p.4, subsection “Model classes”, l.2; p.20, l.24; p.22, subsection “ $\delta^{18}\text{O}$ calculations”, l.1; SI p.2, Suppl. Note 2, l.1; Suppl. Fig. 3, caption, l.5
- There is some ambiguity on the interpretation of benthic $\delta^{18}\text{O}$. Commonly, in terms of Plio-Pleistocene paleoclimate variability, this variable is considered a proxy for the global ice volume and global mean temperature. For interpreting the corresponding long-term variations, especially regarding the “100-kyr problem” and the – according to my best knowledge - still unsettled debate on the origins of the Mid-Pleistocene transition (MPT), one important contribution has been the regolith hypothesis by Clark and Pollard (Paleoceanography, 1998). The latter hypothesis has linked the long-term variations of global ice volume (mainly over the northern hemisphere mid-to-high latitudes) to atmospheric CO₂ levels via regolith erosion by ice sheets. However, this does *not* mean that we should understand benthic $\delta^{18}\text{O}$ variability itself directly as a proxy for CO₂ concentrations and regolith deposits, as suggested by the authors on p.6, ll.3-4. [Interestingly, Clark and Pollard themselves recently admitted that the MPT (in sea-level variability) can be interpreted without their original regolith hypothesis (see, e.g., <https://meetingorganizer.copernicus.org/EGU21/EGU21-13981.html>).] In brief: The considered oxygen isotope ratio should be understood as a proxy for global mean temperature, the variability of which over the Plio-Pleistocene has been primarily determined by the global ice volume. The long-term variability of the latter two climate characteristics has originated from a complex mixture of often nonlinear processes (including CO₂ variations linked to weathering and affecting the global radiation budget on top of orbital parameters’ variations and resulting changes in surface irradiation strength and seasonality), as correctly elaborated by the authors on p.16.
- Similar to my above comment regarding the caption of Fig. 1, I also find the discussion on Fig. 6 on p.11 a bit confusing, starting with the sentence “In particular...” First, the authors report on oscillations on the second half of the central panel, which are hardly visible (very low amplitude?), but on a suppressed

amplitude in the first half (not the second one). Similarly, the authors state that in the lower left panel, the large amplitude is concentrated in the first half of the series, while the figure shows this behavior in the second half only. I strongly recommend revising the two paragraphs below Fig. 6 to establish a clear and unanimous link between what is shown in the figure and what is written in the text.

- P.13, subsection “revealing... trends”: Please clarify if SST and SAT are both indeed averaged *globally*, or SST rather takes only data from (parts of) the Indo-Pacific, as mentioned at other places in the manuscript. (Also in SI, p.1, last line; Suppl. Fig. 2 caption, ll.2 and 3.)
- A few references are incomplete (missing page number or article identifier): refs. 7, 11, 24, 41, 50, 53, 54, 59, 69, 71, 76, 78, and SI ref. 1.
- SI, Suppl. Note 2, l.9: “Myr” should be “kyr”

Technical suggestions:

- P.2, l.19: It might not be clear (at least to me) what the authors mean by “greater cyclicity”. A larger amplitude, a more periodic behavior, or something entirely different? Some minor rewording might help clarifying this point.
- P.3, caption of Fig. 1, l.8: “period of 97.35 time units”
- P.5, caption of Fig. 2: some references to subpanels are shown in italics, others in normal font
- P.6, subsection on Models M & A, l.1: Regarding the reference to Fig. 2 (upper left), I am wondering if the authors mean an embedding in \mathbb{R}^3 (rather than \mathbb{R}^2) or a projection on \mathbb{R}^2 . According to my understanding, a helix is a three-dimensional object.
- P.5, subsection on Model F, ll.2-3: I suggest providing the information that $\mathcal{I}=T/4$ has been used for the embedding of Models M & A already in the corresponding (previous) subsection.
- P.8, l.2: “and a complex-valued function”
- P.8, below Eq. (4): Could you briefly elaborate what “modest K” means here? Which aspects should guide the choice of K in general?
- Figures 4 and 5: I recommend adding a brief information in the respective captions that Fig. 4 (5) refers to Model M (A).
- P.9, l.6 from bottom: “consider P as approximating”
- P.12, subsection on “climate variability...”, end of 2nd paragraph: “...interval spanning a vast part of the industrial era” (130 years is not the entire industrial era). On p.13, l.5, this has been formulated appropriately.
- P.15, l.28: “east of the Andes”
- P.16, ll.4-6: Since some of the orbital parameters vary with more than one frequency, I would suggest speaking of “main periods”.

- Figure 9: It might be nice if the four regions for which the precipitation cycle are shown could be indicated on some map (maybe as boxes in Fig. 8).
- P.17, subsection “Dataset description”: missing space before “oxygen”
- P.22, subsection “ $\delta^{18}\text{O}$ calculations”, l.2: “consists”

I'm looking forward to seeing this interesting work published soon.

Referee’s report on the revision of the paper “Revealing trends and persistent cycles of non-autonomous systems with operator-theoretic techniques: Applications to past and present climate dynamics” by Froyland et al.

The authors provide an application of Koopman and transfer operator frameworks for autonomous systems to analysis of non-autonomous systems. They then apply the framework to two real-world examples from present and past climate dynamics: Variability of sea surface temperature (SST) over the industrial era and the mid-Pleistocene transition (MPT) of Quaternary glaciation cycles. Separating forcing and internal variability aspects is, of course, a valuable endeavor. I thank the authors for the detailed response. However, I am still a bit underwhelmed with the level of the advance.

In the response, the authors state: “We emphasize that the main contributions of the submission do not include the development of new methods to approximate Koopman and transfer operators; indeed the implementations used in the submission have already appeared in the literature [11]. The main novelty lies in the interpretation of the eigenfunctions of stochastically perturbed transfer and Koopman operators, built using standard autonomous approaches from a single nonstationary time series.” But, scanning the literature, autonomous approaches for non-autonomous dynamical systems have been discussed in some detail, and rigorously, e.g. in [1]. In fact, the pitfalls of the approach were pointed out there, and corrective algorithms provided. The examples include those with switching frequency -see section 3.1 - also considered in the current paper. In addition, the type of switching dynamics considered in the current paper was analyzed using “autonomous approaches in [2]. The authors don’t seem to be aware of either of these although both consider nonautonomous systems with frequency switching and single trajectory time series.

An additional, smaller remark is that the authors say they could not recover eigenfunctions of Arbabi-Mazic paper, but I see plot of eigenfunctions in fig 4. of Arbabi-Mezic, so this is confusing.

References

- [1] Senka Macesic, Nelida Crnjaric-Zic, and Igor Mezic. Koopman operator family spectrum for nonautonomous systems. *SIAM Journal on Applied Dynamical Systems*, 17(4):2478–2515, 2018.
- [2] Igor Mezic, Zlatko Drmac, Nelida Crnjaric-Zic, Senka Macesic, Maria Fonoberova, Ryan Mohr, Allan Avila, Iva Manojlovic, and Aleksandr Andrejucuk. A koopman operator-based prediction algorithm and its application to covid-19 pandemic. *arXiv preprint arXiv:2304.13601*, 2023.

Revealing trends and persistent cycles of non-autonomous systems with operator-theoretic techniques: Applications to past and present climate dynamics

Gary Froyland, Dimitrios Giannakis, Edoardo Luna, Joanna Slawinska

March 6, 2024

We thank the Reviewers and Editors for their careful consideration and comments on the manuscript. Below is a point-by-point response to the Reviewers' comments, using an italic font to indicate text copied verbatim from their reports. Changed or added text in the revised manuscript is highlighted in red.

1 Response to Reviewer #1

General and Recommendation. The authors have made a substantial and thorough effort at addressing the issues raised in my first review. The current version is greatly improved and should be published in Nature Communications. The points below are relatively minor and are only meant to improve the final, published paper even more.

We thank the Reviewer for their positive assessment of the revised manuscript.

The authors' effort in Supplementary Note 3 of carrying out a comparison between SSA and their Koopman/transfer operator-based technique is quite praiseworthy. As is often the case in such comparisons, the authors' method wins. In the case at hand, the authors only used the "vanilla" version of SSA, as introduced in the Vautard & Ghil (Physica D, 1989) and Ghil & Vautard (Nature, 1991) papers. In such an implementation, the separation of trend and distinct oscillatory components is not optimized. More recently, Groth & Ghil (PRE, 2011; J. Clim., 2015) have shown that this problem can be fixed by appropriate rotations of the eigenvectors and illustrated the cleaner results thus obtained on very large climatic (Groth et al., J. Clim., 2017) and macroeconomic (Groth & Ghil, Chaos, 2017) data sets. Clearly, I do not insist that the authors turn their attention to these finer points of SSA methodology. But it would be nice to mention that such improvements do exist and provide the appropriate references.

Response: We have added a brief remark on these methodologies along with citations in Supplementary Note 3.

1. It would be good if the authors devised a short name and an acronym for their method, to better place it in the context of other methods.

Response: We agree that it would be nice to have an acronym but since our approach can be implemented using different Koopman/transfer operator approximation methodologies (of which we use two in this paper), we are hesitant to commit to a particular acronym.

2. Likewise, a shorter title for the paper would attract greater attention to the work therein.

Response: Thank you for this suggestion. Upon consideration, we have decided not to shorten the current title as it contains important keywords/phrases for the content of the paper, namely, "trends", "persistent cycles", "non-autonomous systems", "operator-theoretic techniques", and "past and present climate dynamics". We believe that removing any of these would make the title less descriptive.

3. The reference to Mo & Ghil (JAS, 1987) in the current version's ref. [72] is incomplete since it does not include the name of the 2nd author.

Response: Corrected.

4. I'm not sure that I understand the sentence on p. 12, blue para., "This provides an approximate interpretation of the cyclicity of the dominant modes extracted by SSA using many delays." I thought that the embedding in SSA and the authors' method is the same Mañe-Takens embedding.

Response: The embedding is indeed the same for both methods. The statement referred to approximate cyclicity of the eigenfunctions of the covariance operator utilized by SSA. In general, this operator does not commute with the Koopman/transfer operators but asymptotically commutes in the limit of infinitely many delays. In that limit, SSA and Koopman/transfer operators acquire common eigenspaces, and if these eigenspaces are finite-dimensional (which happens by compactness of covariance operators whenever the corresponding eigenvalues are nonzero) they will be spanned by Koopman/transfer eigenfunctions (which are necessarily cyclical for measure-preserving ergodic systems).

5. Slight correction to a sentence on p. 15, 1st para., "These patterns explain a relatively small amount of variance of the raw Indo-Pacific SST data (about two orders of magnitude smaller than the seasonal cycle pair $\{v_1, v_2\}$) and, as we have verified in separate calculations, are not captured in the SSA/EEOF spectrum (possibly since SSA/EEOFs optimize for explained variance [...])" It is only the vanilla version mentioned above that orders EOFs by variance. Neither Monte Carlo SSA (Allen & Smith, J. Clim., 1996) nor the previously mentioned references do.

Response: We now say "possibly since standard SSA/EEOFs optimize for explained variance" to make explicit that our remark refers to vanilla SSA, and refer the reader to Supplementary Note 3 for further discussion.

6. There are several precursors to SSA and multichannel SSA (M-SSA) and we have done our best to mention all those we knew about in the Ghil et al. (Rev. Geophys., 2002) review paper. I wouldn't pick Weare & Nastrom's (MWR, 1982) paper as the most interesting or visionary one of them: their EEOFs were only extended to three successive maps and there was no indication of embedding, determining cyclicity, etc.

Response: We cite Weare & Nastrom as an early EEOF reference.

2 Response to Reviewer #2

The authors provide an application of Koopman and transfer operator frameworks for autonomous systems to analysis of non-autonomous systems. They then apply the framework to two real-world examples from present and past climate dynamics: Variability of sea surface temperature (SST) over the industrial era and the mid-Pleistocene transition (MPT) of Quaternary glaciation cycles. Separating forcing and internal variability aspects is, of course, a valuable endeavor. I thank the authors for the detailed response. However, I am still a bit underwhelmed with the level of the advance.

In the response, the authors state: "We emphasize that the main contributions of the submission do not include the development of new methods to approximate Koopman and transfer operators; indeed the implementations used in the submission have already appeared in the literature [11]. The main novelty lies in the interpretation of the eigenfunctions of stochastically perturbed transfer and Koopman operators, built using standard autonomous approaches from a single nonstationary time series." But, scanning the literature, autonomous approaches for non-autonomous dynamical systems have been discussed in some detail, and rigorously, e.g. in [1]. In fact, the pitfalls of the approach were pointed out there, and corrective algorithms provided. The examples include those with switching frequency -see section 3.1 - also considered in the current paper. In addition, the type of switching dynamics considered in the current paper was analyzed using "autonomous approaches in [2]. The authors don't seem to be aware of either of these although both consider nonautonomous systems with frequency switching and single trajectory time series.

Response: We agree that the switching dynamics considered by Maćesic et al. (2018) [1] is related to Model F studied in the present paper – thank you for bringing this reference to our attention. We note that Maćesic et al. focus explicitly on linear or linearisable non-autonomous dynamical systems. Moreover, they make use of sliding windows for Koopman operator spectral computations which differs significantly from typical autonomous techniques. In contrast, the theoretical setting and typical applications of our framework involve strongly nonlinear dynamics such as climate dynamics. Further, our present work shows that differing frequencies can be separated without needing sliding windows and/or other complicated approaches. We have added a reference to Maćesic et al. in the first paragraph of the introduction and we discuss this paper further in the subsection “Eigenfunctions for Model F”. Regarding the preprint Mezić et al. (2023) [2], their approach also makes heavy use of sliding windows for Koopman operator computations which differs significantly from typical autonomous techniques.

An additional, smaller remark is that the authors say they could not recover eigenfunctions of Arbabi-Mezic paper, but I see plot of eigenfunctions in fig 4. of Arbabi-Mezic, so this is confusing.

Response: We were confused by the notation in that paper. In section 2, their data matrix has dimensions $n \times m$ with m and n corresponding to the number of observations and number of observables, respectively. In section 3, which describes the Hankel-DMD framework, these dimensions are switched around: Their Hankel matrix has dimensions $m \times n$ with m corresponding to the number of delays and n the number of observations. The Hankel-DMD algorithms described in that section reference those of section 2 which led to our confusion. Restoring the correct meaning for m and n , it turns out that the Hankel-DMD implementation and results included in our previous response are consistent with the Arbabi-Mezic paper.

3 Response to Reviewer #3

I have very much enjoyed re-reading this inspiring manuscript after revision. In my opinion, the authors have done a good job in addressing the comments of all reviewers, except for some minor oversights that I will list below along with some additional observations of mine suggesting a few last corrections and clarifications.

We thank the Reviewer for their positive assessment of the revised manuscript.

P.2, ll.24-25: It might help clarifying why the authors suggest a harmonic model to capture both a quite periodic (seasonal) and a rather aperiodic (ENSO) cyclic variation. I suppose that this is just meant as a zero-order “conceptual” model, so why not state this explicitly?

Response: We changed “simple model” to “conceptual model”.

P.3, caption of Fig. 1, ll.10-13: The wording and referencing in those lines is a bit confusing with two sentences of almost identical wording but opposite statements. This may be partially because there is no explicit referencing to the lower panel in the caption. Please revise the caption accordingly.

Response: We are very grateful to the Referee for noticing the missing text in the caption, and for the inconsistencies in pre/post-MPT and fast/slow oscillation. We have added the missing reference to the lower panel and have corrected the caption, also changing the phrasing of the complex eigenfunctions to match the language of the caption of Figure 6.

As already emphasized in my original review, $\delta^{18}\text{O}$ is a ratio of the concentrations of two different oxygen isotopes, not a concentration itself. This is still stated erroneously at a couple of places within the manuscript (and needs corresponding correction): p.4, subsection “Model classes”, l.2; p.20, l.24; p.22, subsection “ $\delta^{18}\text{O}$ calculations”, l.1; SI p.2, Suppl. Note 2, l.1; Suppl. Fig. 3, caption, l.5

Response: Corrected—thank you for catching these omissions.

There is some ambiguity on the interpretation of benthic $\delta^{18}\text{O}$. Commonly, in terms of Plio-Pleistocene paleoclimate variability, this variable is considered a proxy for the global ice volume and global mean temperature.

For interpreting the corresponding long-term variations, especially regarding the “100-kyr problem” and the – according to my best knowledge - still unsettled debate on the origins of the Mid-Pleistocene transition (MPT), one important contribution has been the regolith hypothesis by Clark and Pollard (Paleoceanography, 1998). The latter hypothesis has linked the long-term variations of global ice volume (mainly over the northern hemisphere mid-to-high latitudes) to atmospheric CO₂ levels via regolith erosion by ice sheets. However, this does **not** mean that we should understand benthic $\delta^{18}\text{O}$ variability itself directly as a proxy for CO₂ concentrations and regolith deposits, as suggested by the authors on p.6, ll.3-4. [Interestingly, Clark and Pollard themselves recently admitted that the MPT (in sea-level variability) can be interpreted without their original regolith hypothesis (see, e.g., <https://meetingorganizer.copernicus.org/EGU21/EGU21-13981.html>).] In brief: The considered oxygen isotope ratio should be understood as a proxy for global mean temperature, the variability of which over the Plio-Pleistocene has been primarily determined by the global ice volume. The long-term variability of the latter two climate characteristics has originated from a complex mixture of often nonlinear processes (including CO₂ variations linked to weathering and affecting the global radiation budget on top of orbital parameters’ variations and resulting changes in surface irradiation strength and seasonality), as correctly elaborated by the authors on p.16.

Response: We agree that the text on p. 6 pointed out by the reviewer was inaccurate. We intended to convey that x is a proxy variable for drivers of global mean temperature, which we now state explicitly.

Similar to my above comment regarding the caption of Fig. 1, I also find the discussion on Fig. 6 on p.11 a bit confusing, starting with the sentence “In particular. . .” First, the authors report on oscillations on the second half of the central panel, which are hardly visible (very low amplitude?), but on a suppressed amplitude in the first half (not the second one). Similarly, the authors state that in the lower left panel, the large amplitude is concentrated in the first half of the series, while the figure shows this behavior in the second half only. I strongly recommend revising the two paragraphs below Fig. 6 to establish a clear and unanimous link between what is shown in the figure and what is written in the text.

Response: We are again very grateful to the reviewer for picking up these inconsistencies, which have now been rectified.

P.13, subsection “revealing. . . trends”: Please clarify if SST and SAT are both indeed averaged **globally**, or SST rather takes only data from (parts of) the Indo-Pacific, as mentioned at other places in the manuscript. (Also in SI, p.1, last line; Suppl. Fig. 2 caption, ll.2 and 3.)

Response: The SST and SAT anomalies shown in Figure 7 are indeed based on global averages. This is stated explicitly in the caption on Figure 7 as well as in the first and fourth lines of subsection “Revealing climate change trends through eigenfunctions”. However, the eigenfunctions are always computed using Indo-Pacific SST data. We amended the text in subsection “Dataset and analysis description” to make this more explicit. There, we also say that other variables such as SAT and precipitation fields are not employed in eigenfunction computations. The last two lines of Supplementary Note 1 also make a clear distinction between Indo-Pacific SST data used for transfer operator computations and global SST/SAT time series used in Supplementary Figure 1.

A few references are incomplete (missing page number or article identifier): refs. 7, 11, 24, 41, 50, 53, 54, 59, 69, 71, 76, 78, and SI ref. 1.

Response: Thank you for noticing the missing information. The BibTeX style file we were using (naturemag-doi.bst) did not support an EID field which led to missing article identifiers. The missing information has been added using the pages field.

SI, Suppl. Note 2, l.9: “Myr” should be “kyr”

Response: Corrected.

TECHNICAL SUGGESTIONS:

P.2, l.19: It might not be clear (at least to me) what the authors mean by “greater cyclicity”. A larger amplitude, a more periodic behavior, or something entirely different? Some minor rewording might help clarifying this point.

Response: By having “greater cyclicity” we mean having a well-defined characteristic frequency and slowly decaying correlation amplitude. We now state this explicitly.

P.3, caption of Fig. 1, l.8: “period of 97.35 time units”.

Response: Corrected.

P.5, caption of Fig. 2: some references to subpanels are shown in italics, others in normal font.

Response: Corrected.

P.6, subsection on Models M & A, l.1: Regarding the reference to Fig. 2 (upper left), I am wondering if the authors mean an embedding in \mathbb{R}^3 (rather than \mathbb{R}^2) or a projection on \mathbb{R}^2 . According to my understanding, a helix is a three-dimensional object.

Response: Thank you, it should be \mathbb{R}^3 , and has been corrected.

P.5, subsection on Model F, ll.2-3: I suggest providing the information that $\ell = T/4$ has been used for the embedding of Models M & A already in the corresponding (previous) subsection.

Response: We added this information to the subsection on models M & A.

P.8, l.2: “and a complex-valued function”.

Response: Corrected.

P.8, below Eq. (4): Could you briefly elaborate what “modest K ” means here? Which aspects should guide the choice of K in general?

Response: We have been more specific and indicated why K might need to be larger. There are no hard and fast rules in time-series analysis; generally K should be small to reduce “noise” through perturbation to more distant points, but if K is too small then computing eigenvectors of the matrix may be numerically challenging.

Figures 4 and 5: I recommend adding a brief information in the respective captions that Fig. 4 (5) refers to Model M (A).

Response: This is a good suggestion, we have added an opening sentence in a similar style to that of Figure 6.

P.9, l.6 from bottom: “consider P as approximating”

Response: Thank you, corrected.

P.12, subsection on “climate variability...”, end of 2nd paragraph: “...interval spanning a vast part of the industrial era” (130 years is not the entire industrial era). On p.13, l.5, this has been formulated appropriately.

Response: We changed the text to “an interval spanning the past ~ 130 years of the industrial era.”

P.15, l.28: “east of the Andes”

Response: Corrected.

P.16, ll.4-6: Since some of the orbital parameters vary with more than one frequency, I would suggest speaking of “main periods”.

Response: We changed the text as suggested.

Figure 9: It might be nice if the four regions for which the precipitation cycle are shown could be indicated on some map (maybe as boxes in Fig. 8).

Response: This is a nice suggestion – we’ve added markers in Figure 8 indicating the locations sampled in Figure 9.

P.17, subsection “Dataset description”: missing space before “oxygen”

Response: We checked that the text was typeset correctly.

P.22, subsection “ $\delta^{18}O$ calculations”, l.2: “consists”

Response: Corrected.